# Routing of the RAB6 secretory pathway towards the lysosome related organelle of melanocytes

Anand Patwardhan[1], Sabine Bardin[2], Stéphanie Miserey-Lenkei[2], Lionel Larue[3,4,5], Bruno Goud[2], Graça Raposo[1,6] & Cédric Delevoye[1,6]

Exocytic carriers convey neo-synthesized components from the Golgi apparatus to the cell surface. While the release and anterograde movement of Golgi-derived vesicles require the small GTPase RAB6, its effector ELKS promotes the targeting and docking of secretory vesicles to particular areas of the plasma membrane. Here, we show that specialized cell types exploit and divert the secretory pathway towards lysosome related organelles. In cultured melanocytes, the secretory route relies on RAB6 and ELKS to directly transport and dock Golgi-derived carriers to melanosomes. By delivering specific cargos, such as MART-1 and TYRP2/ DCT, the RAB6/ELKS-dependent secretory pathway controls the formation and maturation of melanosomes but also pigment synthesis. In addition, pigmentation defects are observed in RAB6 KO mice. Our data together reveal for the first time that the secretory pathway can be directed towards intracellular organelles of endosomal origin to ensure their biogenesis and function.

[1] Institut Curie, PSL Research University, CNRS, UMR 144, Structure and Membrane Compartments, Paris F-75005, France. [2] Institut Curie, PSL Research University, CNRS, UMR 144, Molecular Mechanisms of Intracellular Transport, Paris F-75005, France. [3] Institut Curie, PSL Research University, INSERM U1021, Normal and Pathological Development of Melanocytes, Orsay 91405, France. [4] Université Paris-Sud, Université Paris-Saclay, CNRS UMR 3347, Orsay 91405, France. [5] Equipe Labellisée Ligue Contre le Cancer, Orsay 91405, France. [6] Institut Curie, PSL Research University, CNRS, UMR144, Cell and Tissue Imaging Facility (PICT-IBiSA), Paris F-75005, France. Correspondence and requests for materials should be addressed to G.R. (email: graca.raposo@curie.fr) or to C.D. (email: cedric.delevoye@curie.fr).

Eukaryotic cells exploit their intracellular trafficking pathways to exchange molecules between intracellular compartments. Once newly synthesized proteins reach the Golgi apparatus, they can exit at the *trans*-Golgi network (TGN) by sorting into different carriers and transport to various destinations (for example, endosomes, plasma membrane)[1]. Among these multiple routes, the secretory pathway defines a TGN to plasma membrane route, where released post-Golgi carriers are transported along the cytoskeleton and fuse with the cell surface to exocytose their contents[2].

Several components, like RAB GTPases, molecular motors, SNARES or tethering factors cooperate to transport, target and fuse exocytic carriers with the plasma membrane[3]. Among numerous different Golgi-associated small RAB GTPases[4], RAB6 is the most abundant[5] and associated with Golgi cisternae and TGN membranes, as well as vesicles 'en route' to the cell surface. RAB6 modulates the constitutive secretory pathway by controlling the release, motion and docking of secretory carriers with the plasma membrane[6]. Especially, the docking step requires the RAB6-dependent recruitment of co-factors to those carriers, like the RAB6 effector ELKS (refs 6,7) (protein rich in the amino acids E, L, K and S) (a.k.a. RAB6IP2 (ref. 8), CAST2 (ref. 9) or ERC1 (ref. 10)).

In specialized cell types, the RAB6-dependent secretory pathway fulfils specific functions. RAB6 contributes to the immune responses of macrophages or B-lymphocytes by modulating the cell surface delivery of TNF or MHC class II molecules[11,12]. During neuritogenesis, RAB6 controls the transport of exocytic carriers towards the periphery of developing neurons[13]. Similarly, RAB6 supports the polarization of *Drosophila* oocytes by delivering components to cell surface during oogenesis[14,15]. Whereas RAB6 controls the microtubule-dependent anterograde transport of exocytic carriers, ELKS promotes the docking and contributes to the fusion of those vesicles with active zones of the plasma membrane[6] that manifests in a subset of cell types like inhibitory neurons[16,17].

Specialized cell types hosting lysosome-related organelles (LROs) adapt their intracellular trafficking to generate and maintain these particular organelles[18]. Melanosomes are LROs of epidermal and uveal melanocytes (choroid, iris and ciliary body) and of the retinal pigment epithelium (RPE). Their biogenesis relies on the endosomal sorting and trafficking of melanosomal components towards maturing melanosomes[19]. Melanosomes develop in four distinct morphological stages that sequentially mature from unpigmented stages-I/-II to pigmented stages-III/-IV[20]. Most and main constituents of pigmented melanosomes are integral membrane proteins. On one hand, melanogenic enzymes (Tyrosinase (TYR) and related proteins (TYRP1 and TYRP2/DCT)) initiate synthesis of melanins in stage-III, which ultimately fill the lumen of stage-IV; on the other hand, structural proteins like PMEL and MART-1 contribute to the morphology and homoeostasis of melanosomes[19]. Before reaching melanosomes, these neo-synthesized components are sorted at the Golgi and packaged into derived vesicles. However, molecular machineries involved in cargo recognition, sorting, transport and delivery from Golgi are unknown.

By investigating the role of the GTPase RAB6 in melanocytes, we unravel a novel direct Golgi-melanosome route. Unexpectedly, this route shares features of the conventional secretory pathway. RAB6 together with ELKS cooperate to transport, dock and deliver exocytic carriers loaded with specific cargoes (MART-1/TYRP2) to melanosomes. Importantly, this new pathway is required *in vitro*, as well as *in vivo* for melanosome maturation and tissue pigmentation. Altogether, this study uncovers the first evidence that the secretory pathway can be routed towards intracellular compartments and required for their biogenesis and function.

## Results

**RAB6 associates with pigmented melanosomes.** Ubiquitously expressed RAB6 GTPases exist in two alternatively spliced isoforms (RAB6A and RAB6A′, henceforward RAB6) that share numerous effectors and only differ by three amino acids[21]. We first investigated the subcellular localization of RAB6 in melanoma cells (human melanocytic MNT-1 cell line—a well-defined system to study melanosome biology[22,23]), as well as in primary human and mouse melanocytes (NHEM: Normal Human Epidermal Melanocytes; NMM: Normal Mouse Melanocytes). Immunofluorescence microscopy (IFM) showed that endogenous RAB6 localized to the Golgi apparatus and TGN (arrowheads) and to scattered vesicles tightly apposed to TYRP1 and melanin containing melanosomes (Fig. 1a,b and Supplementary Fig. 1a, arrows). An average of 1.5 RAB6-positive fluorescent structures/melanosome was measured in each cell type (Fig. 1c). By live fluorescence imaging of MNT-1 expressing a fluorescent intra-body recognizing endogenous activated RAB6 (AA2-YFP[24]), RAB6:GTP localized to discrete structures associated with melanosomes (positive for mCh-VAMP7 (ref. 25); Supplementary Movie 1), as in fixed cells. Interestingly, some VAMP7-positive melanosomes were fully decorated by RAB6:GTP (Supplementary Fig. 1b, arrow). Similarly by co-expressing VAMP7, TYRP1 or RAB27a as membranous components of pigmented melanosomes, RAB6 fused to GFP or mCherry co-localized extensively with melanosomes (basolateral imaging plane Z-1; Fig. 1d,e and Supplementary 1c-d, arrows; see the Pearson coefficient and the linescan profile) and with the Golgi apparatus (apical imaging plane Z-2; Supplementary Fig. 1d, arrowheads). Only GFP-RAB6 in its active GTP bound form localized to melanosomes since dominant active (Q72L) RAB6 associated to melanosomes, whereas the inactive (T27N) form did not (Supplementary Figs 1e,f, arrows). While the RAB6 localization in melanocytes (endogenous versus exogenous) revealed a difference in its abundance to melanosomes, it indicates that both endo-/exogenous RAB6 associate with melanosomes. Quantitative analyses by immuno-electron microscopy (IEM) revealed three main GFP-RAB6-positive populations morphologically identified as Golgi and TGN (white arrows), small cytoplasmic vesicles (arrowheads) and pigmented melanosomes (stages-III/-IV; black arrows), while pre-melanosomes (stages-I/-II) or other compartments were mostly negative (Fig. 1f,g). Accordingly, endogenous RAB6 and GFP-RAB6, but not GFP alone, associated to the limiting membranes of melanosomes isolated from either NHEM or MNT-1 cells by IEM (Fig. 1h, see Methods), and preferentially to stage-III pigmented melanosomes (Fig. 1i). Lysates from similar melanosomal enriched fractions (Mel) subjected to biochemical analyses were devoid of Golgi (GM130) or cytosolic (GAPDH) components. Those were positive for melanosomal components (TYRP1, TYRP2 and MART-1) and endogenous or exogenous RAB6 (Fig. 1j, stars). Altogether, the results define RAB6, as a melanosome-associated RAB.

**RAB6 controls melanosomes maturation *in vitro* and *in vivo*.** The localization of RAB6 to melanosomes prompted us to address its role during melanosome biogenesis and pigmentation. Depletion of RAB6 (using siRNAs or shRNAs targeting RAB6 isoforms[26]) decreased the melanin content of MNT-1 cells and NHEM by ∼30% from control values (Fig. 2a,b and Supplementary Figs 2a–e). By conventional EM on MNT-1 or NHEM, RAB6 depletion dropped the number of pigmented melanosomes (stages-III/-IV; arrows), but increased the number of unpigmented immature melanosomes (stages-II; arrowheads) (Fig. 2c,d and Supplementary Fig. 2f). We took advantage of the

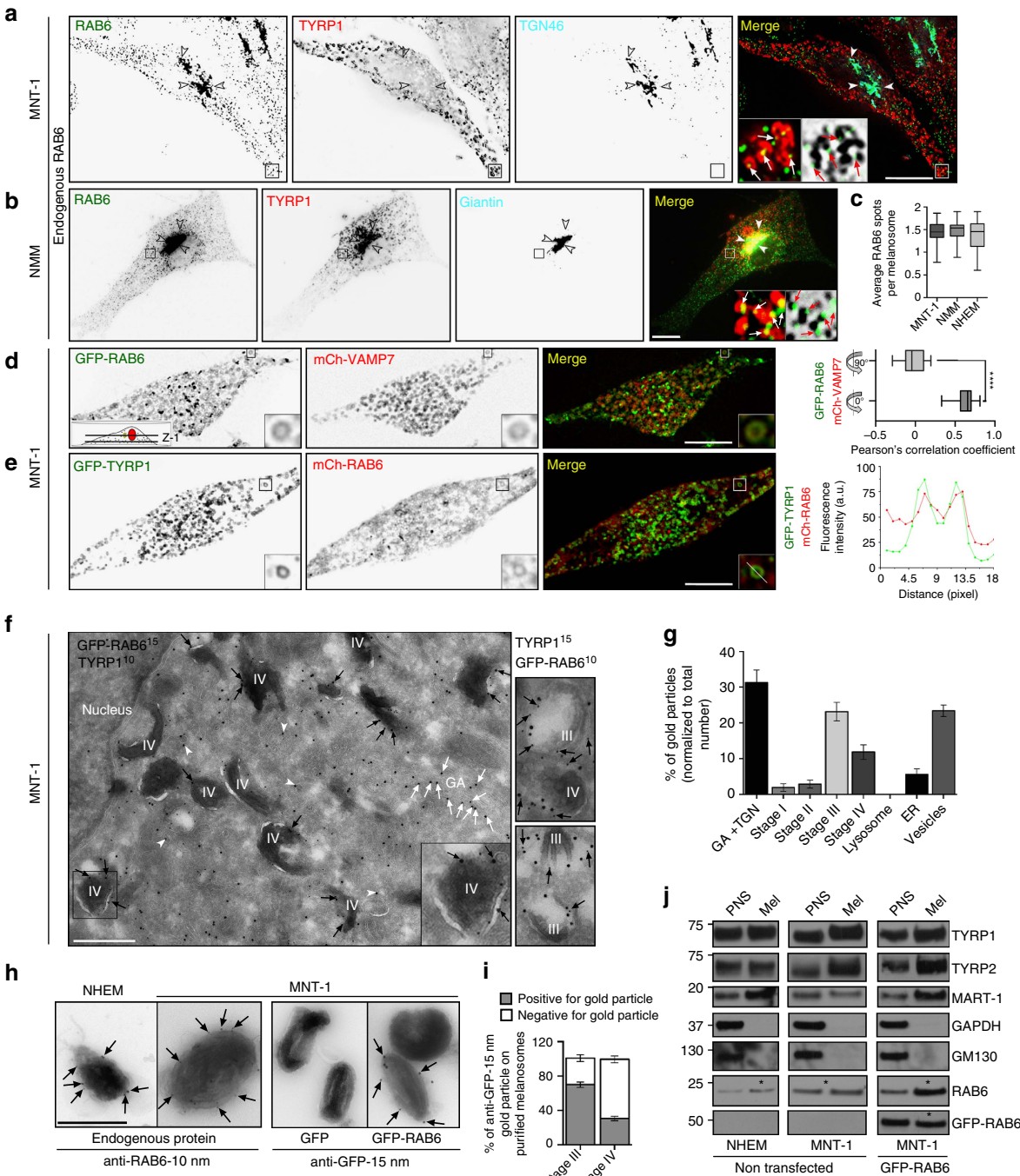

**Figure 1 | RAB6 associates with melanosomes.** (**a,b**) IFM on MNT-1 cells (**a**) and Normal Mouse Melanocytes (NMM) (**b**) stained for endogenous RAB6, TYRP1 and TGN46 (**a**) or Giantin (**b**), showing RAB6 distribution to the Golgi area (arrowheads) and in the vicinity of TYRP1- and melanin-positive melanosomes (white and red arrows respectively, 5X area). (**c**) Quantification of the number of endogenous RAB6 fluorescent structures associated with melanosomes in MNT-1, NMM and NHEM. (**d**) Live imaging frames of MNT-1 (n) cells (imaging plane Z-1) expressing GFP-RAB6 and mCh-VAMP7 (Pearson's correlation coefficient; n = 66, four independent experiments). (**e**) Live imaging frames of MNT-1 cells co-expressing GFP-TYRP1 and mCh-RAB6 with corresponding intensity profile of RAB6 across melanosomes measured along white line (right panel). (**f**) Representative electron micrograph of ultrathin cryosection of GFP-RAB6 expressing MNT-1 cells showed RAB6 association (anti-GFP, × 2 area) to Golgi Apparatus (GA, white arrows), vesicles (arrowheads) and limiting membranes of melanosomes (black arrows) positive for TYRP1. (**g**) Quantification of immunogold labelling for GFP-RAB6 relative to indicated cell compartments (230 gold particles; n = 12 cells, 2 independent experiments). Data are normalized to the total number of gold particles and presented as means ± s.e.m. (**h**) Immunogold labelling of RAB6 (PAG-10 nm) or GFP (arrows) of melanosome-enriched fractions from NHEM (left) or MNT-1 expressing GFP or GFP-RAB6 (PAG-15 nm, right). (**i**) Distribution of anti-GFP gold particles over stage-III/-IV purified melanosomes from MNT-1 expressing GFP-RAB6 presented as mean ± s.e.m. (n = 209 melanosomes, with at least 1 gold particle, quantified from 2 independent experiments). (**j**) Western Blot (WB) of PNS (1%, Post-nuclear supernatant) and melanosomes-enriched fraction (25%, Mel) isolated from NHEM or MNT-1 cells expressing or not GFP-RAB6 and probed with RAB6 or GFP (*), TYRP1, TYRP2, MART-1, GM130 or GAPDH antibodies. Blots are representative of 3 independent experiments. Scale bars: 10 μm (**a,b** and **d,e**), 500 nm (**f–h**), M.W.in kDa, two-tailed, Unpaired t test, ****P < 0.0001.

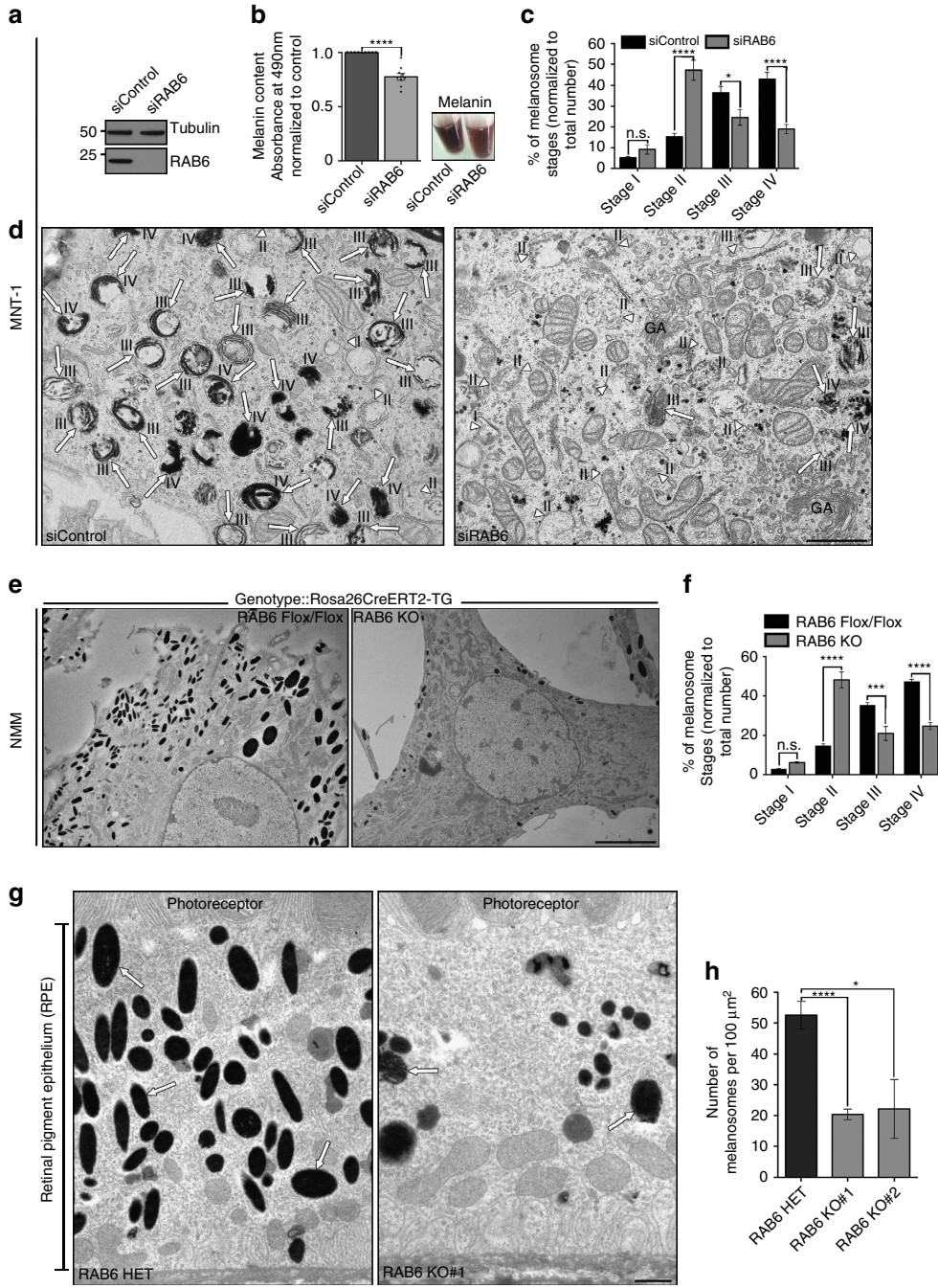

**Figure 2 | RAB6 controls the biogenesis of melanosomes *in vitro* and *in vivo*.** (**a**) WB of control (siControl) or RAB6 (siRAB6) depleted MNT-1 cell lysates probed with RAB6 or Tubulin antibodies. (**b**) Melanin content estimation of siControl or siRAB6 depleted MNT-1 cells (9 independent experiments). (**c**) Quantification of immature (stages-I/-II) and mature (stages-III/-IV) melanosomes (n), presented as mean percentage of the total number ± s.e.m. (siControl, $n = 725$; siRAB6, $n = 395$, 8–10 cells per condition, four independent experiments). (**d**) Conventional EM analysis of control or RAB6-depleted MNT-1 cells (stages-I/ -II, arrowheads; stages-III/ -IV, arrows). (**e**) Conventional EM on NMM isolated from *Rosa26CreERT2- TG$^{-/+}$ or $^{+/+}$; Rab6$^{F/F}$* mouse treated (RAB6 KO) or not (RAB6 Flox/Flox; control) with 4-OHT to extinguish RAB6 expression. (**f**) Quantification of immature (stages-I/ -II) and mature (stages-III/-IV) melanosomes (n) in NMM. Data are means ± s.e.m. (RAB6 Flox/Flox, $n = 1468$; RAB6 KO, $n = 826$; 4 independent mice). (**g**) Conventional EM analysis of RPE from 2 independent conditional RAB6 KO (*Tyr::Cre/°; Rab6$^{F/F}$*) mice showed a reduction in pigmented melanosomes (arrows) relative to RAB6 HET mice (*Tyr::Cre/°; Rab6$^{F/+}$*). (**h**) Quantification of the number of pigmented melanosomes (n) per 100 μm$^2$ of RPE in RAB6 HET and conditional RAB6 KO mice. Data are means ± s.e.m. (RAB6 HET, $n = 526$; RAB6 KO#1, $n = 74$; RAB6 KO#2, $n = 204$) from 10 random electron micrographs. Scale bars: 500 nm (**d**), 5 μm (**e**), 1 μm (**g**); M.W. in KDa; Two tailed, Unpaired *t* test (**b,h**), Sidak's multiple comparisons test (**c**); *$P < 0.05$; **$P < 0.01$; ***$P < 0.001$; ****$P < 0.0001$.

recently generated conditional RAB6A knockout mice (Rosa26CreERT2-TG) to isolate primary melanocytes for which RAB6A and RAB6A′ expression was extinguished on incubation with 4-hydroxy tamoxifen (4-OHT)[27]. Similarly to human

melanocytes, endogenous RAB6 localized specifically to discrete structures (∼1.5/melanosome) closely associated with pigmented melanosomes (arrow). These structures disappeared upon five days incubation with 4-OHT (Supplementary Fig. 2g,h), as

confirmed by IFM and immunoblotting (Supplementary Fig. 2i,j). RAB6 KO melanocytes analysed by EM revealed similarly an increase of stage-II unpigmented melanosomes associated to a concomitant drop of pigmented stage-III/-IV melanosomes relative to controls (Fig. 2e,f and Supplementary Fig. 2k). This result strongly supports a role for RAB6 during the biogenesis (stages-II to -III) of pigmented melanosomes.

*In vivo*, RPE expresses TYR and represents an excellent model for melanosome biogenesis[28]. Using adult mice in which RAB6 was specifically knocked out (KO) in the melanocyte lineage (*Tyr::Cre*/°; *Rab6^F/F* (RAB6 KO) or *Tyr::Cre*/°; *Rab6^F/+* (RAB6 HET (Heterozygote) conditional mice; Supplementary Fig. 2l and see also Methods), the number of pigmented melanosomes (arrows) was dramatically decreased in RPE from RAB6 KO mice relative to RAB6 HET RPE (Fig. 2g,h). Altogether, our results show that RAB6 is critical for the biogenesis and pigmentation of melanosomes *in vitro* and *in vivo*.

**RAB6 vesicles dynamically contact melanosomes**. We next examined how RAB6 functions to control melanogenesis. For that purpose we have followed the dynamics of RAB6-positive vesicles relative to melanosomes by live imaging on MNT-1 co-expressing mCh-VAMP7 (melanosomal associated SNARE[29]) and GFP-RAB6. In agreement with other studies[6,30,31], GFP-RAB6-positive vesicles emerged and moved away from the Golgi (Supplementary Movie 2, top panel) along microtubules (Supplementary Movie 2, bottom panel, arrowhead). As already observed for activated RAB6 (Supplementary Movie 1), these Golgi-derived RAB6 vesicles encountered directly and sequentially distinct melanosomes with which they established dynamic contacts (Supplementary Movie 3, arrowheads). Manual tracking of GFP-RAB6 vesicles revealed that most of them underwent anterograde movements defined by an average velocity of $0.88 \pm 0.04\,\mu\mathrm{m\,s^{-1}}$ (Fig. 3a and Supplementary Movie 4, see also Supplementary Table 1). Interestingly, many of those vesicles (84% of the total tracked vesicles, arrows) acted differently once in the vicinity of VAMP7-positive melanosomes (arrowheads) by slowing down until a complete pause (average pause duration) lasting $5.4 \pm 0.6\,\mathrm{s}$ (Fig. 3b, Supplementary Fig. 3a and Supplementary Movie 5), suggesting a specific docking of RAB6 vesicles to melanosomes.

**Secretory cargos are delivered to melanosomes**. To investigate whether the association of RAB6-positive vesicles with melanosomes precedes a fusion step, we followed in live cells the trafficking of a secretory cargo, NPY (Neuropeptide Y), known to be transported in post-Golgi RAB6-positive vesicles[6]. NPY-Venus co-distributed with endogenous activated RAB6 (AA2) at the Golgi and on peripheral vesicles (Supplementary Fig. 3b,c, arrowheads and arrows, respectively; Pearson's coefficient, $0.61 \pm 0.02$), and as confirmed by IEM (Fig. 3c, arrows). Similar to RAB6 vesicles, NPY-positive vesicles were closely associated to pigmented melanosomes (Fig. 3d, arrows). However and in contrast to RAB6, NPY accumulated within the lumen of melanosomes (Fig. 3e–g, arrows) instead of their limiting membranes (Fig. 1f,h, black arrows). In addition, fusion events defined by the partial exchange of NPY fluorescence from the vesicle to the melanosome lumen (Fig. 3h,i, arrows and Supplementary Movie 6) were observed. These results suggest that post-Golgi derived RAB6 vesicles loaded with exocytic cargoes (NPY) can fuse with melanosomes, allowing RAB6 to associate with the limiting membrane of melanosomes and soluble cargoes to be released into melanosomes. Fusion occurs most likely via a kiss and run process because one RAB6-positive vesicle contacts melanosomes (Supplementary Movies 3 and 5) without being consumed.

On the other hand and as observed in HeLa cells[6], MNT-1 cells secreted a large amount of NPY-Venus (Supplementary Fig. 3d), indicating that the conventional RAB6-dependent secretory pathway between Golgi and the plasma membrane is still functional. Altogether, these results suggest that, in melanocytes, the RAB6-dependent secretory pathway is adapted and partly re-routed towards the melanosomes.

**ELKS cooperates with RAB6 during melanosome biogenesis**. ELKS is a RAB6 effector that functions along the secretory route by promoting the docking of secretory vesicles to spatially organized plasma membrane areas[6,32,33]. To investigate whether ELKS also functions in the Golgi-melanosome transport pathway, we compared the pigmentation status and melanosome biogenesis in MNT-1 cells depleted of RAB6 or ELKS alone or together. Although MNT-1 cells highly expressed ELKS as compared to others cells[8], ELKS siRNAs treatment resulted in a ∼50% decrease of ELKS without affecting RAB6 or VAMP7 expression (Fig. 4a and Supplementary Fig. 4a). Spectroscopy analysis showed that depletion of RAB6, ELKS or both decreased the intracellular melanin content (∼25% relative to controls; Fig. 4b). In addition, conventional EM on single- or double-depleted cells showed a decreased number of pigmented melanosomes (stages-III/-IV, arrows) concomitantly to an increase of immature melanosomes (stages-I/-II, arrowheads) relative to controls (Fig. 4c,d). The lack of synergistic effect following depletion of RAB6 and ELKS suggests that they function in the same pathway.

We then assessed the localization of endogenous or over-expressed ELKS (in fusion with GFP) in MNT-1 cells. Endogenous ELKS localized to punctate structures close to plasma membrane (Supplementary Fig. 4b, left panel, arrowheads and as observed in other cell types[6]) and to pigmented melanosomes (here labelled with CD63 (ref. 34)) (Fig. 4e; arrows). These ELKS-positive structures in the vicinity of melanosomes were lost upon staining with pre-immune serum (Fig. 4e, left) or ELKS depletion, as illustrated by the dramatic decrease in ELKS fluorescent intensity (Fig. 4e,f, right panel). However, ELKS-positive structures were partially preserved at the plasma membrane of depleted cells (Supplementary Fig. 4b; arrowheads). This suggests the existence of two pools of ELKS with different half-lives; one stable and most likely anchored at the plasma membrane, and a second much more dynamic (and affected by siRNA depletion) closely associated to melanosomes.

By IEM, GFP-ELKS co-localized with TYRP1 to melanosomes and was tightly associated with electron-lucent vesicles (Fig. 4g; arrows and arrowheads, respectively). Accordingly, endogenous ELKS was detected together with RAB6 in pigmented melanosomal fractions (asterisk) devoid of Golgi (GM130) or cytoskeleton (tubulin) contaminants, but positive for melanosomal components (TYRP1, MART-1 or RAB27) (Fig. 4h). Importantly and in addition to its basolateral localization at plasma membrane (arrowheads), GFP-ELKS co-distributed also with mCh-RAB6-positive vesicles (Fig. 4i, arrows) emerging out of the Golgi area (Supplementary Movie 7, top panel, arrowheads) or with RAB6:GTP (AA2) in fixed cells (Supplementary Fig. 4c, arrows). However, a C-terminal deleted version of ELKS lacking its RAB6 binding domain, GFP-ELKS-ΔCt[6,8], did not co-distribute with mCh-RAB6-positive vesicles (Fig. 4i, arrows and Supplementary Movie 7, bottom panel), while preserving its basal plasma membrane localization (Fig. 4i, arrowheads). This proposes that RAB6 recruits ELKS on budding vesicles at TGN or soon after their egress[6,35]. Lastly, cells expressing GFP-ELKS-ΔCt failed to localize GFP-RAB6 to donut-like melanosomes as compared to GFP-ELKS-expressing MNT-1 (Fig. 4j). These data suggest that ELKS cooperates with RAB6 during melanosome biogenesis by

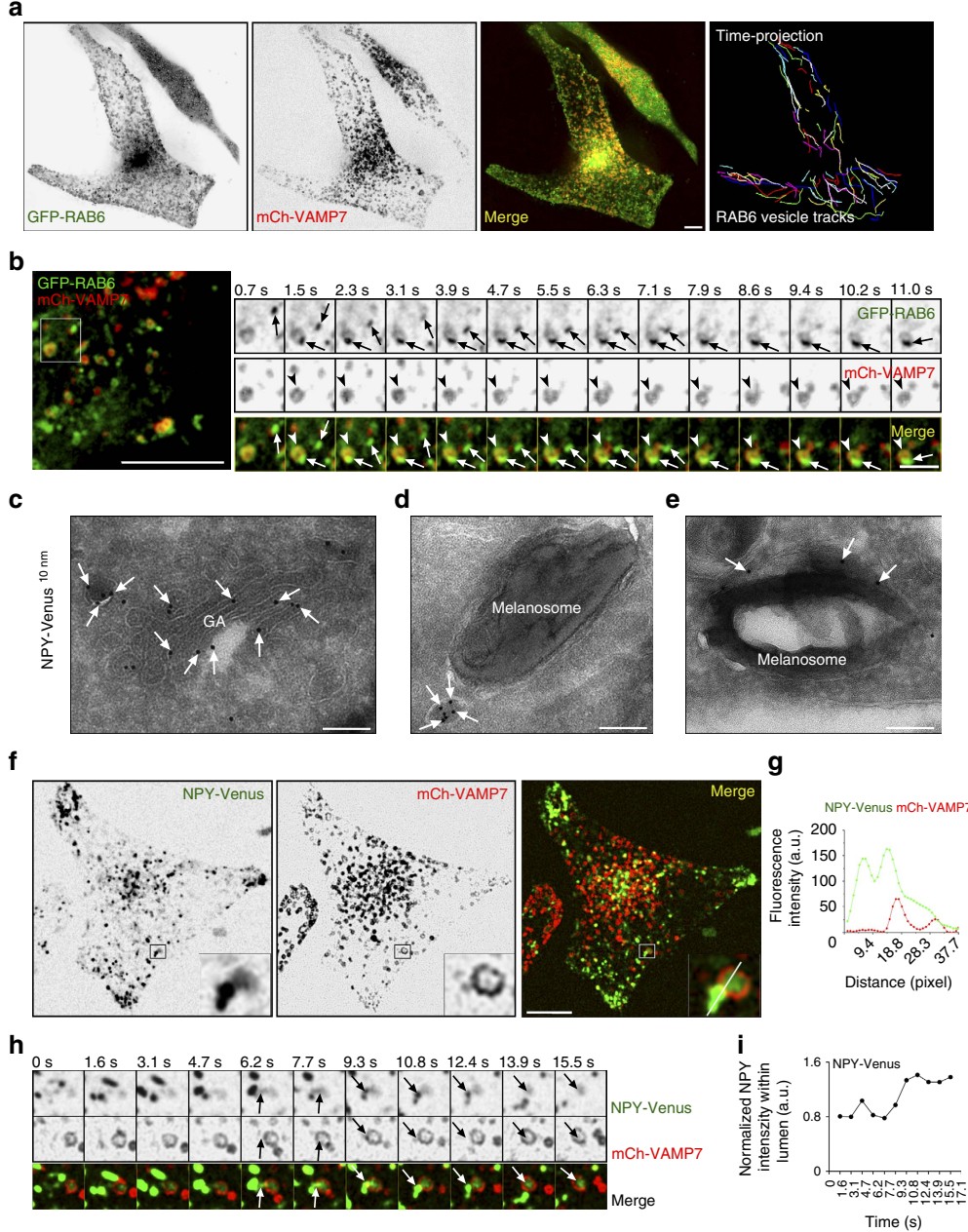

**Figure 3 | RAB6 vesicles fuse with melanosomes and deliver secretory cargo into melanosomes.** (**a**) Live imaging frame of MNT-1 cells expressing GFP-RAB6 and mCh-VAMP7 and corresponding time projection of RAB6-positive vesicles tracks (right) (see Supplementary Movie 4). (**b**) Consecutive time-lapse images of GFP-RAB6 vesicles (arrows) approaching and docking to GFP-RAB6 and mCh-VAMP7 positive melanosomes (arrowheads, and Supplementary Movie 5). (**c**–**e**) Electron micrographs of ultrathin cryosections of NPY-Venus expressing MNT-1 cells showed NPY association (PAG-10 nm, arrows) to Golgi apparatus (GA) and derived vesicles (**c**), as well as vesicles nearby melanosomes (**d**) and also in the lumen of melanosomes (**e**). (**f**) Live imaging frame of MNT-1 cells expressing NPY-Venus and mCh-VAMP7. (**g**) Linescan intensity profiles of NPY across VAMP7-positive melanosomes measured along white lanes highlighted by the 5X area in **f**. (**h**) Consecutive time-lapse images of the insets (in **f**) showing the partial delivery of NPY cargoes from the vesicle to the lumen of the melanosome (arrows and Supplementary Movie 6). (**i**) The NPY-Venus fluorescent intensity within mCh-VAMP7 melanosomes (in **h**) was measured over time and normalized to the mCherry intensity. Scale bars, 5 μm (**a,b,f**); 100 nm (**c**–**e**).

associating with post-Golgi derived vesicles 'en route' to and fusing with melanosomes.

**ELKS targets RAB6 vesicles to melanosomes.** ELKS is thought to specify docking sites of RAB6-positive secretory vesicles with the plasma membrane[6]. We thus investigated whether ELKS fulfils the same function at the melanosomal membrane. ELKS depletion in MNT-1 cells did not affect the Golgi distribution of RAB6 (arrowheads), as well as the number of dynamic RAB6-positive vesicles[6] (Fig. 5a,b). However and similarly to GFP-ELKS-ΔCt expressing cells (Fig. 4j), ELKS-depleted cells exhibited a two-fold reduction in the number of cells displaying GFP-RAB6 as a donut-like melanosomal structure (Fig. 5a,c, arrows). In addition, GFP-RAB6 co-localization with mCh-VAMP7 was dropped relative to controls (Fig. 5a,d, arrows). These results suggest that the fusion of RAB6-positive vesicles with melanosomes requires the expression of ELKS and its interaction with RAB6.

The dynamic of GFP-RAB6 vesicles in ELKS-depleted cells displayed a similar average velocity to siCTRL or non siRNA-treated cells (Fig. 5e; see also Supplementary Table 1). Yet ELKS-depleted cells showed a ~30% reduction in the number of GFP-RAB6 vesicles undergoing pauses nearby melanosomes (Supplementary Table 1) as compared to control. Although the tracking time of GFP-RAB6 vesicles was similar in control- and ELKS-depleted cells (Fig. 5f), the total pause fractions of the tracks decreased by ~50% (Fig. 5g and Supplementary Table 1). Indeed, fewer RAB6-positive vesicles were immobilized upon ELKS depletion and especially in the vicinity of melanosomes (Supplementary Table 1), indicating that ELKS is involved in the docking of RAB6 vesicles with melanosomes.

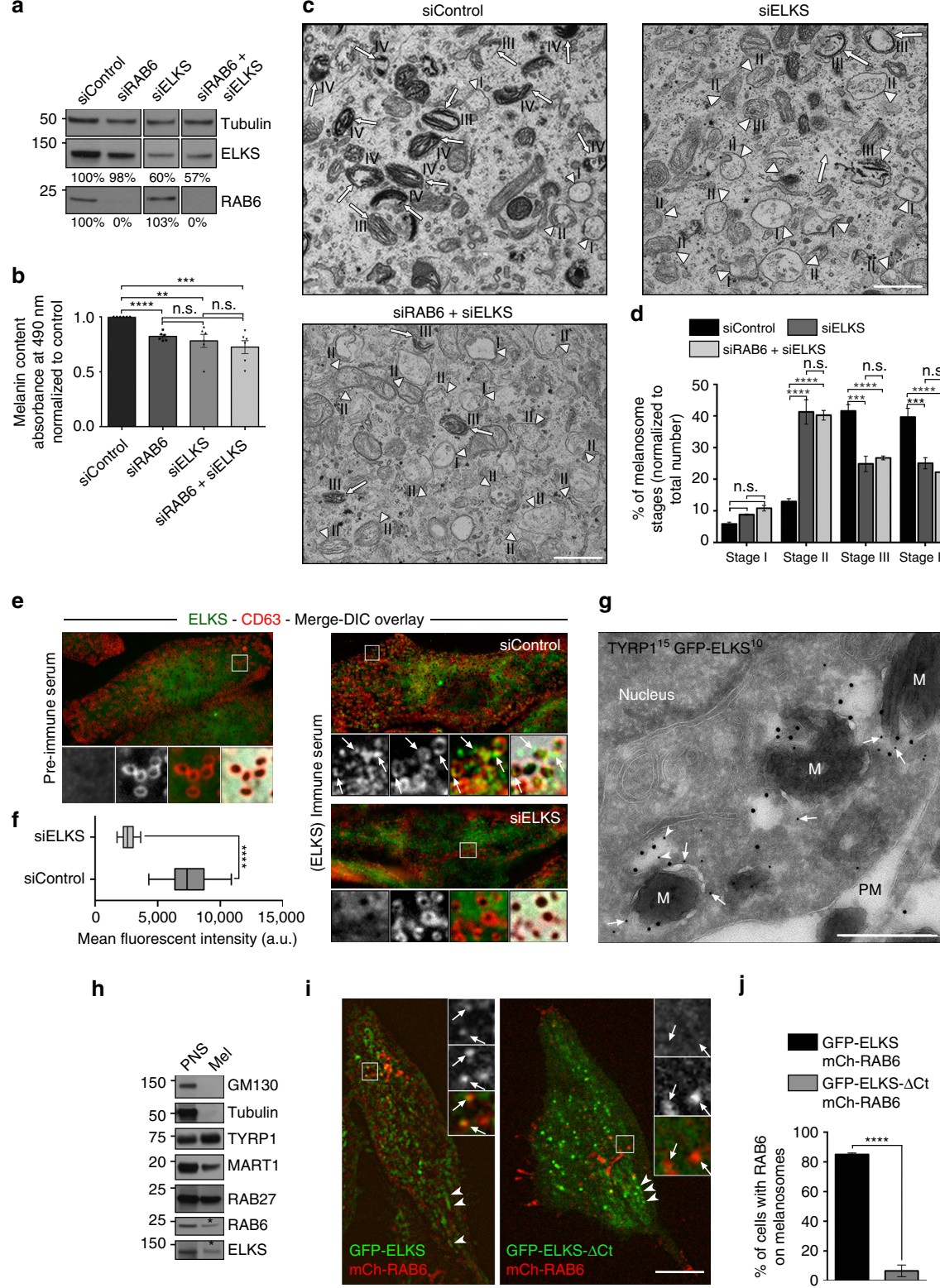

**Melanosomal cargoes follow the secretory pathway**. RAB6 and ELKS define a novel Golgi to melanosome route required for the biogenesis of melanosomes. Thus, we investigated which melanosomal cargoes could follow this route and contribute to pigmentation. Biochemical analysis of RAB6-depleted cell lysates of MNT-1 or NHEM showed a two-fold increase in the expression of specific trans-membrane cargoes like TYRP2 and MART-1, while TYR, TYRP1 or PMEL was unchanged (Fig. 6a,b and Supplementary Fig. 5a–c). Neither the migration profiles on SDS gels of these heavily glycosylated melanosomal cargoes nor the post-Golgi processing of PMEL[36] were affected (Supplementary Fig. 5a–c), indicating no overall alteration of the Golgi function (glycosylation or transport). However, biochemical association of MART-1 and TYRP2 to melanosome-enriched fractions was decreased (55.1 ± 14% and 45.6 ± 14.5%, associated with $P$ values 0.0327 and 0.0197, respectively and normalized to TYRP1, Fig. 6c,d; asterisks), while TYRP1 remained stable. These results strongly suggest a role for RAB6 in the specific Golgi-to-melanosome delivery of MART-1 and TYRP2.

While well studied melanogenic cargoes, such as TYR, TYRP1 or PMEL, travel through early endosomes before their sorting and transport to melanosomes[19,23,36], the intracellular trafficking of MART-1 or TYRP2 remains unknown. We thus investigated the localization of those two cargoes relative to RAB6 in NHEM, MNT-1 or HeLa cells. Due to the lack of a suitable TYRP2 antibody for immuno-detection, a chimera consisting in the fusion of GFP with the trans-membrane and cytosolic tail domains of human TYRP2 (GFP-TYRP2) was generated and co-distributed with mCh-VAMP7 to melanosomes (arrows), but not with transferrin-positive endosomes (Supplementary Fig. 5d, top panel). Similar results were obtained with endogenous MART-1 (Supplementary Fig. 5d, bottom panel, arrows). In addition to stages-II and -III melanosomes, endogenous MART-1 was previously localized on small and electron dense Golgi-derived vesicles[37]. Actually, a majority of the GFP-RAB6-positive vesicles were co-stained for MART-1 (64%, 74 vesicles over 5 Golgi area), which co-localized almost totally with TYRP2 (105 vesicles over 11 Golgi area) (Fig. 6e,f, black arrows; and co-distributed on Golgi apparatus, white arrows). This suggests that MART-1 and TYRP2 are packaged into the same RAB6-positive vesicles. We thus investigated whether RAB6 functions in the trafficking of endogenous human MART-1 to melanosomes. Apart from its overall increased expression (Fig. 6a,b and Supplementary Fig. 5a), but decreased association with melanosomes (Fig. 6c,d), depleting RAB6 in NHEM or MNT-1 cells resulted in MART-1 accumulation at the TGN (Fig. 6g,h and Supplementary Fig. 5e, arrows). However depleting RAB6 did not affect the localization of TYRP1, which still associated with remaining pigmented melanosomes in MNT-1 (Supplementary Fig. 5f, arrows). This indicates that MART-1 requires specifically RAB6 to exit the Golgi area and to reach melanosomes.

To determine if MART-1 represents a canonical cargo of the RAB6-dependent trafficking pathways, we used HeLa cells as an heterologous system. On its overexpression, MART-1-GFP localized on dynamic peripheral mCh-RAB6 vesicles (arrowheads) and cell periphery (arrows) (Supplementary Fig. 5g and Supplementary Movie 8) and at the extracellular side of the plasma membrane (Supplementary Fig. 5h, left panel), suggesting that MART-1-GFP traffics along the secretory pathway. Interestingly, IFM on non-permeabilized RAB6- or ELKS-depleted HeLa cells showed a loss of MART-1 at the cell surface relative to controls (Supplementary Fig. 5h,i). In addition, ELKS depletion induced the scattering of the residual exocytosed MART-1 (Supplementary Fig. 5h, arrows, right panel), indicating that ELKS might promote the docking of MART1-positive vesicles at specific membrane sites. Altogether, these results unravel MART-1 and most likely TYRP2 as cargoes that follow the Golgi-melanosome RAB6/ELKS-dependent secretory pathway.

## Discussion

Transmembrane cargoes follow the secretory pathway to reach the plasma membrane for further association or secretion, but whether this route targets also intracellular compartments is not known. Here, we reveal that pigment cells redirect the RAB6 secretory pathway towards their LROs and exploit this novel Golgi-to-LRO route for the biogenesis of melanosomes and the pigmentation process. Our results illustrate that this pathway requires the same molecular machinery as reported for the conventional secretory route, namely RAB6 and ELKS. While RAB6 participates in the formation and transport of Golgi-derived vesicles loaded with melanogenic cargoes, ELKS contributes to the recognition of melanosomes as a docking and fusion site for these secretory vesicles.

Melanosomes are the melanocyte's LRO that originate and receive components from the endocytic pathway[18]. Even if suggested[20,38,39], the Golgi apparatus has never been defined as a direct source organelle for incoming melanosomal components. Furthermore, such Golgi-to-LRO route might be shared by other cells generating LROs like platelets and megakaryocytes in which RAB6 co-fractionate with α-granules[40] or antigen presenting cells that require RAB6 for MHC class II molecules cell surface delivery[11].

The secretory pathway defines a Golgi to plasma membrane pathway that require RAB6 and its effector ELKS to dock exocytic vesicles to specific areas[6]. We demonstrate that RAB6 and ELKS associate with melanosomes and equally contribute to their maturation. Moreover, a well-known RAB6- and ELKS-dependent

**Figure 4 | ELKS associates and cooperates with RAB6 during the biogenesis of melanosome.** (**a**) WB of MNT-1 cell lysates depleted for RAB6, ELKS or both probed with respective antibodies and tubulin as loading control. (**b**) Melanin content estimation of cells treated with indicated siRNAs. Data are normalized to control and presented as means ± s.e.m. (six independent experiments). (**c**) Conventional EM analysis of control, ELKS or RAB6 + ELKS-depleted MNT-1 cells (stages-I/-II, arrowheads; stages-III/-IV, arrows). (**d**) Quantification of immature (stages-I/-II) and mature (stages-III/-IV) melanosomes (n) presented as mean percentage (normalized to total n) ± s.e.m. (8–10 cells per condition, three independent experiments. siControl, $n = 572$; siELKS, $n = 342$; siRAB6 + siELKS, $n = 347$). (**e**) IFM on siControl (top) and siELKS (bottom) MNT-1 cells stained for endogenous ELKS using pre- (left) or -immune (right) sera that label specific structures associated to CD63-positive pigmented melanosomes (arrows). (**f**) Quantification of the total ELKS fluorescent intensity measured at the melanosomal plane in control and ELKS-depleted MNT-1 cells. (**g**) Ultrathin cryosections of GFP-ELKS expressing MNT-1 cells co-stained for TYRP1 (PAG-15 nm) and GFP (PAG-10 nm), showed ELKS localization to the limiting membranes of melanosomes (M), (arrows) and associated vesicle (arrowhead). (**h**) WB of PNS (1%) and melanosomes-enriched fraction (25%, Mel) isolated from MNT-1 cells and probed for RAB6 and ELKS (*), TYRP1, MART-1, RAB27, GM130 or GAPDH. (**i**) Live imaging frames of MNT-1 co-expressing GFP-ELKS or GFP-ELKS-ΔCt with mCh-RAB6. GFP-ELKS, but not GFP-ELKS-ΔCt, co-distributed with mChRAB6-positive vesicles close to the Golgi area (arrows, × 5 zoomed area, and Supplementary Movie 7). (**j**) Quantification of the number of cells (n) harbouring GFP-RAB6 to melanosomes (GFP-ELKS, $n = 117$; GFP-ELKS-ΔCt, $n = 84$). Data are normalized to control and presented as mean ± s.e.m. (3 independent experiments). Scale bars: 1 µm (**c**); 10 µm (**e,i**); 500 nm (**g**); M.W. in KDa; two tailed, Unpaired $t$ test (**b,f**), two way ANOVA multiple comparisons (**d**); **$P < 0.01$; ***$P < 0.001$; ****$P < 0.0001$; n.s., non significant.

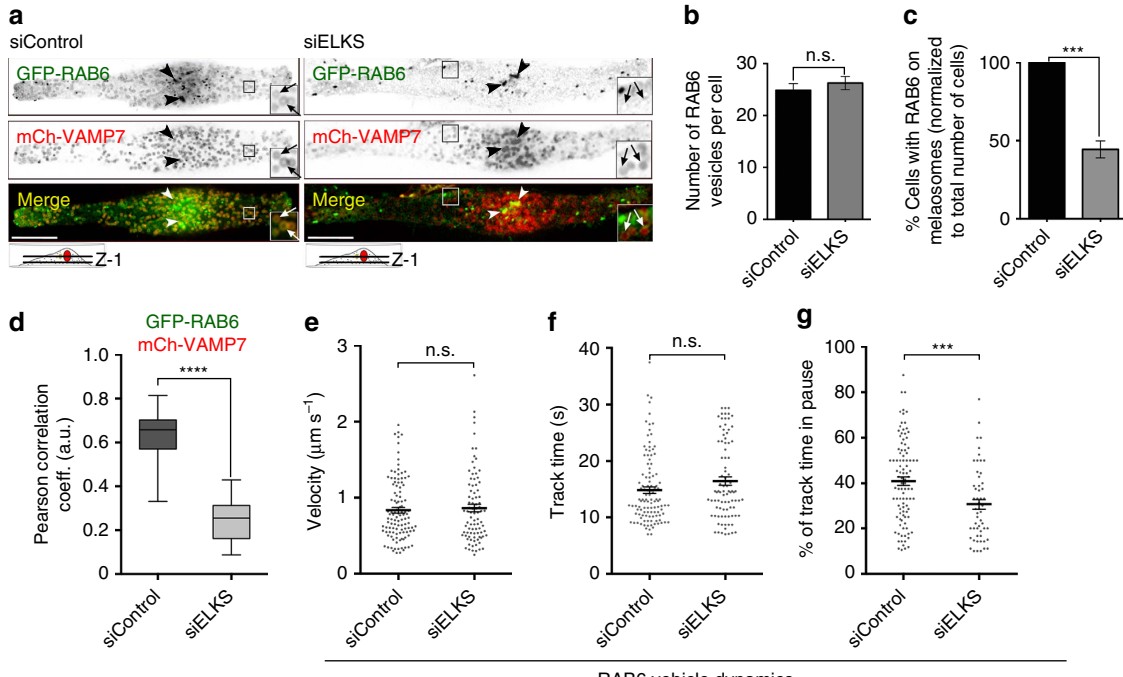

**Figure 5 | ELKS targets RAB6 vesicles to melanosomes.** (**a**) Live imaging frame (basolateral imaging plane Z-1) of control- or ELKS-depleted MNT-1 cells expressing GFP-RAB6 and mCh-VAMP7. Localization of GFP-RAB6 to melanosomes (arrows), but not to Golgi area (arrowheads), was specifically lost upon ELKS depletion ( × 2.5 area). (**b**) Number of RAB6 vesicles quantified per cell (*n*) on live imaging frames in control- and ELKS-depleted MNT-1 cells presented as mean ± s.e.m. (*n* = 14–16, 3 independent experiments). (**c**) Quantification of the number of cells (*n*) harbouring GFP-RAB6 to melanosomes (siControl, *n* = 105; siELKS, *n* = 128). Data are normalized to control and presented as mean ± s.e.m. (three independent experiments). (**d**) Pearson's correlation coefficient of green (GFP-RAB6) and red (mCh-VAMP7) signal intensities in Control- (67 cells) or ELKS- (24 cells) depleted cells (four independent experiments). (**e**–**g**) Characterization of the dynamics of GFP-RAB6-positive vesicles in control- or ELKS-depleted MNT-1 cells. Comparison of velocity (**e**), track time (**f**) and fraction of track time in pause (**g**) presented as scattered plots (Mean ± s.e.m.; see Supplementary Table 1). Scale bars, 10 µm (**a**); two tailed, Unpaired *t* test (**b**–**g**); ***P < 0.001; ****P < 0.0001; n.s., non significant.

secretory cargo (NPY)[6], exogenously expressed in melanocytes, accumulate in the lumen of melanosomes upon partial fusion of RAB6 vesicles. This illustrates that the secretory pathway is routed towards a new target organelle: the melanosome.

Whether melanocytes have developed two distinct secretory routes ending either to melanosomes or plasma membrane or have exploited the constitutive pathway by diverting it to melanosomes is not clear. As observed in HeLa cells[6,7], depleting ELKS or impairing its binding to RAB6 in melanocytes does not impact the formation of RAB6 vesicles and their transport to the cell periphery, but decreases their pause duration close to melanosomes, as documented for the plasma membrane[6]. Also, melanocytes secrete a large amount of NPY, showing that the RAB6 conventional secretory pathway is functional in those cells. But in melanocytes, and in contrast to the constitutive secretory pathway[6,7], RAB6 operates for the sorting and/ or exit of melanosomal cargoes (MART-1 and TYRP2) in Golgi-derived vesicles to which ELKS binds in a RAB6-dependent manner. Then, RAB6 and ELKS control their specific delivery to melanosomes, whereas the RAB6/ELKS couple promotes an efficient spatio-temporal secretion in non-pigment cells[6,7]. Using live cell imaging, RAB6 vesicles emerging from the Golgi area are directly transported towards melanosomes. However and due to their fast motions and the numerous peripheral melanosomes, we failed to discriminate whether those vesicles can also reach the plasma membrane, and thus whether they constitute one or two sub-populations of RAB6 Golgi-derived carriers. However by sharing some features with the secretory pathway, this novel Golgi-melanosome route

corresponds most likely to an adapted and distinct version of the constitutive pathway.

A direct TGN to late endosome pathway transporting lysosomal LAMP1/2 cargoes was recently unravelled in non-pigment cells[41]. Whether this pathway requires RAB6 or ELKS has not been investigated. However and even if MART-1 and TYRP2 contain tyrosine-based sorting motifs (YXXØ) analogous to LAMP1/2 (ref. 42), exogenous MART-1 and most likely TYRP2 reach the plasma membrane of HeLa cells rather than lysosomes, as demonstrated for TYR or TYRP1 (refs 43–45). This observation suggests that MART-1 and TYRP2 use specifically the secretory pathway (as revealed by the lack of overlap with internalized transferrin) and do not require solely their sorting YXXØ determinants to be packaged into exocytic carriers.

What makes the melanosome suitable for fusion with RAB6 secretory carriers? Upon ELKS depletion or impairment of the ELKS-RAB6 interaction, RAB6 vesicles establish fewer contacts with melanosomes, likely decreasing the fusion rates and the association of RAB6 to melanosomal membranes; altogether indicating that ELKS is pivotal for targeting RAB6 vesicles to melanosomes. In non-pigment cells (HeLa cells or hippocampal neurons), ELKS localizes to plasma membrane and/or exocytic vesicles[35] and is a regulatory component that restricts the fusion of RAB6 vesicles to specialized areas of the plasma membrane[6,16,17]. In melanocytes, ELKS localizes to focal adhesion-like structures[46], but also to RAB6 vesicles and melanosomes, indicating that ELKS might be stabilized at melanosome membranes to constrain RAB6 vesicles to fuse with melanosomes.

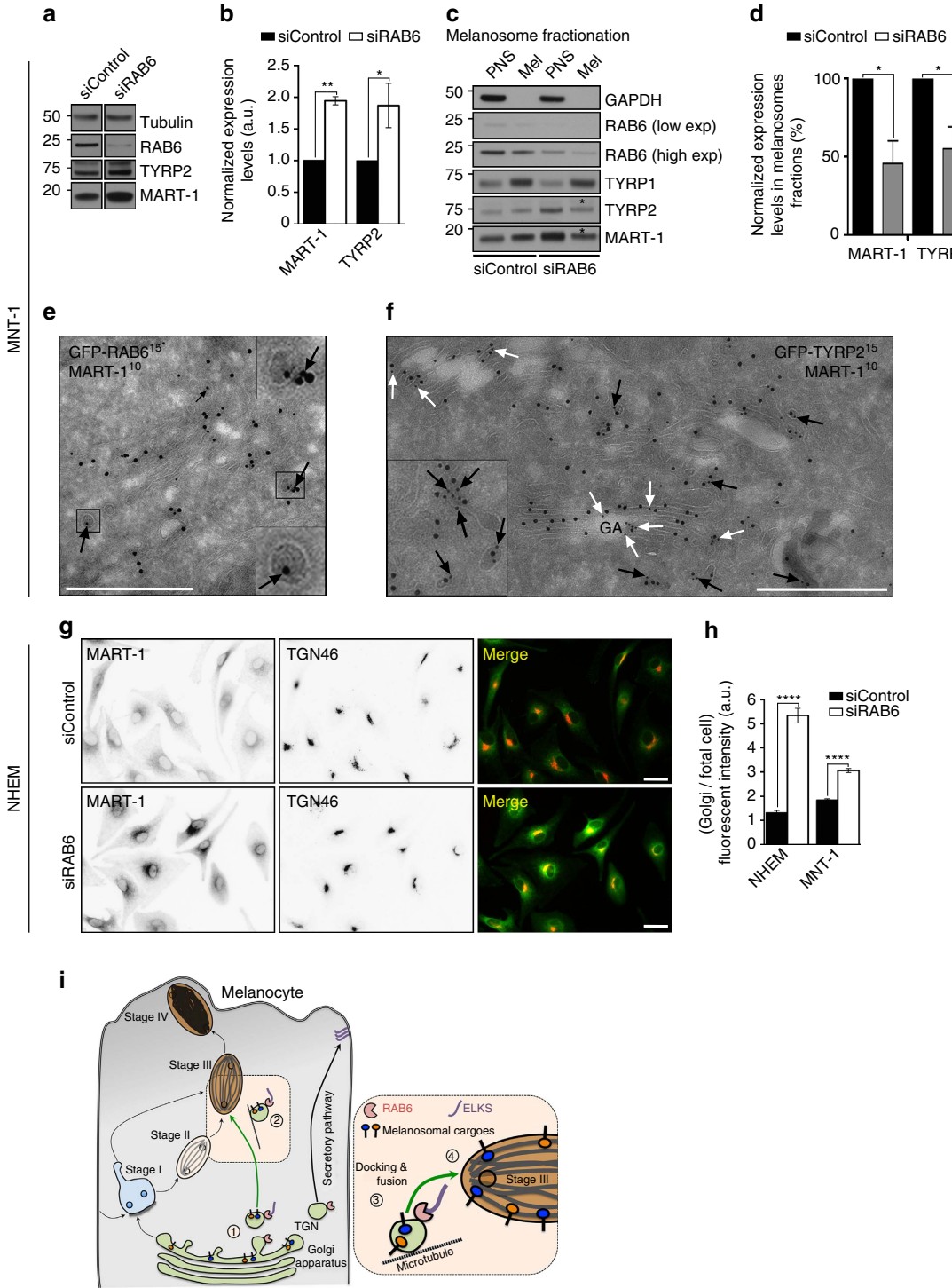

**Figure 6 | RAB6 controls the trafficking of melanosomal cargoes.** (**a**) WB of control or RAB6-depleted MNT-1 cell lysates probed for RAB6, TYRP2, MART-1 or Tubulin. (**b**) Quantification of the relative expression levels of MART-1 or TYRP2 in control or RAB6 siRNAs-treated cells. Data are normalized to control and presented as mean ± s.e.m. (n = 7, TYRP2; n = 3, MART-1). (**c**) WB of PNS (1%) and melanosomes-enriched fraction (5%, Mel) isolated from control- or RAB6-depleted MNT-1 cells and probed for RAB6 (*) and GAPDH (left panel) or TYRP1, TYRP2 and MART-1 (right panel). Note the increase in TYRP2 and MART-1 levels in PNS (black asterisk) and corresponding decrease in Mel fractions (white asterisk). (**d**) Quantification of the expression level of MART-1 and TYRP2 in melanosomes-enriched fractions of RAB6-depleted melanocytes relative to controls. Data are normalized presented as mean ± s.e.m. of three independent experiments. (**e,f**) Immunogold labelling on ultrathin cryosections of GFP-RAB6 (d) or GFP-TYRP2 (e) expressing MNT-1 cells co-stained for GFP (PAG-15 nm) and MART-1 (PAG-10 nm) (arrows). Black arrows point co-distribution of MART-1 and TYRP2 on similar vesicles as in (**d**), and white arrows on Golgi apparatus (GA). (**g**) IFM on control- or RAB6-depleted NHEM stained for endogenous MART-1 and TGN46. (**h**) Quantification of the MART-1 fluorescent intensity enrichment in the Golgi area in control- or RAB6-depleted NHEM or MNT-1 cells (n) presented as mean ± s.e.m., (NHEM, n = 51–63, MNT-1, n = 22–24, per respective condition, three independent experiments). (**i**) Working model illustrating the possible role of RAB6 and ELKS in the Golgi to melanosomes pathway. Note that other alternative routes cannot be excluded. Scale bars: 500 nm (**d**, **e**), 20 μm (**f**). M.W. in kDa, two-tailed, Unpaired t test (**b,c,g**). *P < 0.05; **P < 0.01; ****P < 0.0001.

Several studies revealed that RAB6 controls the release of vesicles from the TGN[27,47] through its cooperation with myosin II and the actin cytoskeleton[30]. As a consequence, loss of RAB6 decreases the surface expression of secretory cargoes like VSV-G[30] or secretion reporter molecules[7]. Knocking-down RAB6 in melanocytes results in the accumulation of MART-1 in the Golgi area, while decreasing its association to melanosomes (as TYRP2), and its exocytosis at the plasma membrane of HeLa cells. This suggests that one function of RAB6 in melanocytes is likely to release vesicles loaded with melanosomal cargoes from the TGN before their microtubule-dependent transport to melanosomes.

While TYR or TYRP1 follow an endosome-to-melanosome route, how TYRP2 or MART-1 are addressed to melanosomes are unknown[19]. By directing Golgi-derived vesicles towards maturing melanosomes, RAB6 and most likely ELKS define a secretory route used by specific cargoes like MART-1 and TYRP2 to reach melanosomes, allowing their maturation and melanocyte pigmentation. While MART-1 might function as chaperon molecule, as shown for PMEL[48], TYRP2 would aid melanin biosynthesis[49,50]. These novel findings lead us to propose the following working hypothesis for melanosome biogenesis (Fig. 6i). RAB6 could recruit ELKS to TGN budding vesicles loaded with MART-1 and TYRP2, promote their release (1) and their transport along the microtubule network (2). ELKS could associate with and contribute to the specific docking of RAB6 vesicles to melanosomes (3). Then RAB6 vesicles would fuse in a 'kiss and run' mechanism with maturing melanosomes and release their contents. This would allow the delivery of transmembrane melanosomal cargoes to the limiting membrane of melanosomes (4). Consequently, this novel pathway co-exists with the well-defined endosome-to-melanosome route and controls melanosomes biogenesis and melanocyte pigmentation.

Using adult mice, where RAB6 expression is specifically lost in the melanocytic lineage, we observed a pigmentation defect in RPE cells. Even if the transition between immature unpigmented and mature pigmented melanosomes cannot be addressed in vivo at the adult stage[28], our results indicate strongly that this secretory pathway is also exploited and is required in vivo for eye pigmentation. Altogether, our study reveals that melanocytes direct the secretory pathway towards melanosomes to ensure their maturation and function. Given that cells hosting LROs share multiple pathways and molecular machineries required for LRO biogenesis[18], the adaptation of the RAB6-ELKS secretory pathway might be a common feature exploited by specialized cell types to fulfil their particular functions.

## Methods

**Antibodies and plasmids.** _Immunofluorescence._ Sheep anti-TGN46 (1:400; AbD Serotec, AHP500GT), rabbit Anti-GFP (1:200; Invitrogen, A-11122), mouse anti-TYRP1 (1:200, Abcam, clone TA99), rabbit anti-RAB6 (1:1,000; Bruno Goud, Institut Curie, Paris France), rabbit anti-ELKS; pre-immune/immune serum (1:50; Bruno Goud, Institut Curie, Paris France), human monoclonal anti-RAB6: GTP[24] (1:100; AA2), anti-Giantin (1:200) and anti-Human Cy3 (1:200; Institute Curie, Recombinant Antibodies Platform), mouse anti-CD63 (1:200, Abcam, CLB-180, ab23792), rabbit anti-MART-1 (1:200; Abcam, EP1422Y), mouse anti-MART-1 (1:200; Thermo fisher scientific, M2-7C10), anti–rabbit, -mouse, -sheep conjugated to Alexa Fluor 488, 555 or 647 (1:200; Invitrogen).

_Western blot._ Rabbit anti-GM130 (1:10,000; Abcam, ab52649), -TYRP1 (1:1,000; Santa Cruz, sc-25543), -GAPDH (1:10,000; Sigma, G9545), -β-tubulin (1:1,000; Abcam, ab6046), -RAB6 (1:1,000 Santa Cruz, sc-310), -MART-1 (1:200; Abcam, ab51061); Mouse anti-Actin (1:1000; Sigma, A5316), -MART-1 (1:200; Thermo fisher scientific, M2-7C10); -Transferrin receptor (1:1,000; Thermo fisher scientific, H68-4), -Tyrosinase (1:200; Santa Cruz, sc-20035), -TYRP2 (1:200; Santa Cruz—sc-74439); Rat anti-GFP (1:1,000; Chromoteck, 3H9); Goat anti-mouse or –rabbit coupled to horseradish peroxidase (HRP) (1:10,000; Abcam, ab6721).

_Electron microscopy._ Rabbit Anti-GFP (1:100 Invitrogen, A-11122), mouse anti-TYRP1 (1:40, abcam, TA99), mouse anti-MART-1 (1:40; Thermo fisher scientific, M2-7C10), rabbit anti-RAB6 (1:40; Bruno Goud, Institut Curie,

Paris France), protein A conjugated to 10/15 nm gold particles (1:50; Cell Microscopy Center, Utrecht University Hospital, Utrecht, Netherlands).

_Plasmids._ encoding GFP/mCh-RAB6A, -RAB6A', GFP-RAB6-Q72L, -T27N and NPY-Venus (Bruno Goud, Institut Curie, CNRS-UMR144, Paris France); AA2-YFP (Franck Perez, Institut Curie, CNRS-UMR144, Paris France); GFP-ELKS and GFP-ELKS-ΔCt (Anna Akhmanova, Utrecht University, The Netherlands); mCherry-VAMP7 (Thierry Galli, INSERM ERL U950, Institut Jacques Monod, Université Paris 7); GFP-TYRP1 (Michael S. Marks, Children's Hospital of Philadelphia, Dept of Pathology & Laboratory Medicine, Philadelphia, PA, USA); GFP-RAB27a (Clotilde Théry, Institut Curie, U932, Paris, France); untagged MART-1 (Donata Rimoldi, University of Lausanne, Switzerland); mCherry-β-Tubulin (Anne Paoletti, Institut Curie, CNRS-UMR144, Paris France); GFP-TYRP2 and MART-1-GFP clones were constructed by PCR amplifying the genes from cDNAs synthesized from MNT-1 cells, and cloning into pEGFP-C3 and pEGFP-N1, respectively, using Gibson Assembly.

**Sequences for RNAs interference.** The sense strand for the indicated double-stranded siRNAs were synthesized with the following sequences or derived from the following references: siRNA control, 5′-AATTCTCCGAACGTGTCACGT-3′ (Qiagen), siRNA RAB6 (targeting RAB6A/A'): 5′-GACATCTTTGATCACCAGA-3′ (Sigma), ELKS siRNA: 5′-GTGGGAAAACCCTTTCAAT-3′ (ref. 46). Rab6 and 26 s RNA Real time PCR Primers were described previously[51,52]; shRNA RAB6#1: GTGCAGGCAAGGCACTGTAAA, shRNA RAB6#2: AGGAGGCTGTTCCT-GCTAATC, shRNA scrambled were a kind gift of Dr Subba R.G. Setty (IISc, Bangalore, India).

**Cell culture and transfections.** MNT-1 cells were cultured in DMEM supplemented with 20% FBS (Invitrogen), 10% AIM-V medium (Life Technologies), non-essential amino acids, sodium pyruvate and antibiotics[20], tested free for mycoplasma contaminations and originally obtained from Dr. Michael S. Marks (Children's Hospital of Philadelphia, Philadelphia, USA). Human foreskin neonatal melanocytes (normal human epidermal melanocytes, NHEM) were obtained from Cellsystems (Cat 104-05n) and cultured in DermaLife Basal Medium with DermaLife M Life Factors[51]. Primary mouse melanocytes were derived and cultured using established procedure[53]. Briefly, skin melanocytes were explanted from 4 to 5 days old mice and cultured in Ham's F-12 Nutrient Mix (GlutaMAX supplement; GIBCO) supplemented with 10% FBS, 1% antibiotics and 200 nM phorbol 12-myristate 13-acetate (TPA; Sigma)[54,55]. For the induction of Rab6 depletion, primary cells were treated with ethanol (untreated; control) or 4-OH tamoxifen (500 nM) for 5 days. For siRNAs knockdown experiment, $1 \times 10^6$ cells were seeded in 10-cm dish on day 1 and transfected on day 3 (200 pmoles) according to the manufacturer's protocol (Oligofectamine; Invitrogen). On day 5, cells were transfected again with similar method, collected and processed for experiments on day 7. MNT-1 or HeLa cells were transfected with plasmids using Lipofectamine 2000 (Thermo Fischer Scientific) or jetPRIME (Polyplus), respectively[22,56]. Lentiviral particles for shRNA infection experiments were produced as previously described from 293FT cells[57]. Briefly, $0.8 \times 10^6$ cells per well were seeded in a 6 well plate in DMEM, 10% FBS (GIBCO) supplemented with antibiotics. The cells were transfected the same day with TransIT respecting the following quantities per well: 8 μl of TransIT-293 (Euromedex), 0.4 μg of pCMV-VSV-G, 1 μg of psPAX2 and 1.6 μg of pLKO.1-Puro-shScramble or pLKO.1-Puro-shRAB6#1 or #2. The cells were left overnight with the transfection mix. The medium was then replaced with 3 ml of RPMI medium with Glutamax (GIBCO), 10% FBS (GIBCO), PenStrep (GIBCO), Gentamicin (50 μg ml⁻¹, GIBCO) and HEPES (GIBCO). The supernatant was then harvested after 32 h, filtered on a 0.45 μm filter, and used to infect MNT-1 cells in presence of 8 μg ml⁻¹ of Protamine (SIGMA). The cells were then selected with 4 μg ml⁻¹ of Puromycine two days after transduction for 7 days.

**Mice breeding and genotyping.** _Mice breeding._ The generation of _Rab6a_ knockout mice is described in ref. 27. Adult animals used in this study were obtained from natural mating between C57BL/6J _Tyr::Cre/°_; _Rab6F/+_ (RAB6 HET) males and C57BL/6J _Tyr::Cre/°_; _Rab6F/+_ (RAB6 HET) females, to generate 25% of progeny with a _Tyr::Cre/°_; _Rab6F/F_ (RAB6 KO) genotype, 25% of progeny with a _Tyr::Cre/°_; _Rab6+/+_ (RAB6 WT) genotype and the other 50% with a _Tyr::Cre/°_; _Rab6F/+_ (RAB6 HET) genotype. The other animals used were obtained from natural mating between C57BL/6J _Rosa26CreERT2-TG⁻/+_; _Rab6F/F_ males and C57BL/6J _Rosa26CreERT2-TG⁻/+_; _Rab6F/F_ females, to generate 1/3 of progeny with a _Rosa26CreERT2-TG⁻/⁻_; _Rab6F/F_ (RAB6 Flox/Flox even after treatment with 4-OH tamoxifen) genotype, 1/3 of progeny with a _Rosa26CreERT2-TG⁻/+_; _Rab6F/F_ (RAB6 KO after treatment with 4-OH tamoxifen) genotype and the other 1/3 with a _Rosa26CreERT2-TG⁺/+_; _Rab6F/F_ (RAB6 KO after treatment with 4-OH tamoxifen) genotype. Animals were maintained according to European and national regulation (approval number of the establishment Trouillet C75-05-18). It complies also with internationally established principles of replacement, reproduction and refinement in accordance with the guide for the care and use of laboratory animals (NRC 2011).

_Mice genotyping._ Genomic DNA was extracted from ear of mice with lysis buffer (Tris–HCl 50 mM pH8, NaCl 100 mM, Tween20 0.5% and 2 mg ml⁻¹ of

proteinase K), and genotyped by PCR using the following primers: 5′- TTGCCT-CCCTGTTTGTACCAGTACGCT -3′(ol1) and 5′- CTTCAACACAAGCCATGA-AGGATCTGG 3′(ol2), which target the sequences before and after the floxed exon4 of the *Rab6a* gene, allowing to distinguish between *Rab6*$^{+/+}$ (RAB6 WT) (231 bp) and *Rab6*$^{F/+}$ (RAB6 HET) alleles (362 bp).

**Biochemistry.** For melanosome purification, NHEM or MNT-1 cells (80% confluence in 10-cm dish) were collected 1 day later by trypsinization, centrifuged (1,200 r.p.m., 5 min) and pellets were suspended in Buffer-I (25 mM HEPES, 1 mM EDTA, 0.1 M EGTA, 0.02% Sodium Azide, pH7.4) with 0.25 M sucrose and homogenized using dounce homogenizer. Lysates were centrifuged (600g, 10 min, 4 °C) to clear off heavy material, and Post-Nuclear Supernatant (PNS) were loaded on Buffer-I containing 2 M sucrose and centrifuged (11,000g, 30 min, 4 °C). Melanosome-enriched fractions (Mel) were gently transferred in different ultracentrifuge tubes and pelleted (55,000g, 60 min, 4 °C). The pellet was lysed using western blot buffer (50 mM Tris, 150 mM NaCl, 10 mM EDTA, 0.1% Triton-X-100, pH 7.4), followed by protein estimation (Pierce BCA Protein Assay Kit). Then equal amount of protein lysates were loaded on 4–12% SDS–PAGE (NuPAGE, Invitrogen) for western blot analyses[23]; uncropped scans of the most important blots were presented (Supplementary Fig. 6). All molecular weights (MW) were in kDa. For melanin assay, cells were disrupted by sonication in 50 mM Tris–HCl, pH 7.4, 2 mM EDTA, 150 mM NaCl, 1 mM dithiothreitol, (with protease inhibitor cocktail, Roche). Pigment was pelleted at 20,000g for 15 min at 4 °C, rinsed once in ethanol/ether (1:1), and dissolved in 2M NaOH/20% dimethylsulfoxide at 60 °C. Melanin content was measured as optical density at 490 nm (Spectramax 250, Molecular Devices).

**Fluorescence imaging.** MNT-1 cells seeded in 6-well plates on coverslips were subjected to siRNA knockdown as described above. Cells were fixed using 2% PFA for all studies unless mentioned. For endogenous ELKS labelling, cells were fixed in 100% ice-cold methanol ( − 20 °C, 5 min) followed by incubation in PBS with 1 mg ml$^{-1}$ BSA (blocking buffer (BB)) all along the procedure. For endogenous MART-1 labelling, cells were fixed in 0.25% Triton X-100, 2% PFA final concentration (90 s), followed by 2% PFA fixation (10 min), followed by incubation in BB all along the procedure. For surface labelling of GFP-MART-1, live HeLa cells were incubated with rabbit anti-MART-1 antibody (1:200) in 1% BSA-DMEM (4 °C, 45 min), fixed directly with 2% PFA (10 min) and processed for IFM by incubation in BB all along the procedure. For intracellular IF, fixed cells were washed in PBS, neutralized (10 min; 50 mM glycine in PBS), saturated in BB, and permeabilized in PBS, 0.05% saponin, and 1 mg ml$^{-1}$ BSA (incubation buffer (IB)). Cells were incubated with the primary antibody diluted in IB (45 min), washed three times in IB, and incubated with the corresponding secondary antibody (30 min). Cells were washed twice in IB and once in BB. Finally, coverslips were mounted in DABCO medium and examined under a 3D deconvolution microscope (DM-RXA2; Leica) equipped with a piezo z drive (Physik Instrument) and a 100 × 1.4 NA Plan Apo objective lens for optical sectioning. 3D multicolour image stacks were acquired using MetaMorph software (MDS Analytical Technologies) through a cooled charge-coupled device (CCD) camera (Coolsnap HQ; Photometrics). All IFM images were deconvoluted.

For time-lapse microscopy, MNT-1 cells were grown on fluorodish (World Precision Instruments) with culture medium supplemented with 10 mM Hepes. Time-lapse imaging was performed at 37 °C (Life Imaging Services) using a spinning-disk microscope mounted on an inverted motorized microscope (TE2000-U; Nikon) through a 100 × 1.4 NA Plan Apo objective lens. The apparatus is composed of a spinning-disk head (CSU-22; Yokogawa), a laser lounge (with a 491-nm Cobalt for GFP observation; Roper Industries), a CCD camera (Coolsnap HQ2; Photometrics) for image acquisition, and MetaMorph software to control the setup. The acquisition parameters used were 100 to 500-ms exposure for GFP and mCherry channels. Transferrin uptake protocol has been described elsewhere[23,56]. Briefly, MNT-1 cells were starved in DMEM (30 min) and incubated 30 min with 10 µg ml$^{-1}$ Tf-A555 or − A647 at 37 °C, fixed with 2% PFA and stained for respective antibodies[23,56].

**Conventional- and immuno-electron microscopy.** *Conventional EM on cells.* MNT-1, NHEM or NMM cells grown on cover slips, and transfected with specific siRNAs or treated with 4-OHT, were fixed with a mixture of 2% (wt/vol) paraformaldehyde, 1% (wt/vol) glutaraldehyde in 0.2 M phosphate buffer (PB), pH 7.4, post-fixed with 1% (wt/vol) OsO4 supplemented with 1.5% (wt/vol) potassium ferrocyanide, dehydrated in ethanol, and embedded in epon[20]. Ultrathin sections were prepared with a Reichert UltracutS ultramicrotome (Leica), counter-stained with uranyl acetate and lead citrate, and viewed with a Transmission Electron Microscope (TEM; Tecnai Spirit G2; FEI, Eindhoven, The Netherlands) equipped with a 4k CCD camera (Quemesa, Olympus, Muenster, Germany).

*Conventional EM on tissues.* RAB6 HET and RAB6 KO animals were sacrificed and the eyes were immediately procured by enucleation and fixed in 2.5% Glutaraldehyde, 2% PFA, 0.1 M Cacodylate pH 7.2, 0.05% CaCl$_2$ for 2 h at 4 °C (ref. 58). The cornea was cut off and the lens was removed. Then the eye cup was post-fixed in 1% osmium, 1.5% potassium ferrocyanide in 0.1 M cacodylate for 2 h, 4 °C. After dehydration in alcohol, the eye cups were transferred to propylene

oxide/CY212 Epon (1:1) before embedding[58]. Ultrathin sections (70–80 nm) were prepared and observed as described above.

*Immunogold labelling.* For ultrathin cryo-sectioning and immunogold labelling, cells were fixed with a mixture of 2% PFA and 0.2% glutaraldehyde in 0.1 M phosphate buffer, pH 7.4. Cells were processed for ultra-cryo-microtomy and single or double-immunogold labelling using protein A conjugated to 10 nm gold (PAG10) or 15 nm gold (PAG15). For immunogold labelling on purified melanosomes, melanosome enriched fraction was purified, as described above. The pellet (55,000g, 60 min, 4 °C) was suspended in PBS, and 4 µl were loaded on EM grid, fixed with 2% PFA and processed for immune-gold labelling using anti-GFP antibody, with secondary PAG15. Electron micrographs were acquired under electron microscopes, as described above.

**Image analysis and quantitations.** Endogenous RAB6 fluorescent structures (number and intensity) in proximity with melanosomes were quantified using ICY software (Plugin Spot Detector) by delineating melanosomes with circles (15 pixels diameter)—which overlap the TYRP1 labelling at melanosomes. Linescan analysis was performed using MetaMorph. Pearson's correlation coefficient between green and red channels intensities (GFP-RAB6/mCh-VAMP7 and NPY-Venus/ RAB6-AA2) was calculated using the JACoP plugin of ImageJ FIJI[59]. As controls, measures were performed on the same images, for which one channel was rotated by 90°, as described earlier[60]. For Pearson's correlation coefficient between NPY-Venus and RAB6-AA2, both live cell imaging frames and acquisition on fixed samples were taken into consideration. Fluorescent intensity was measured on the whole cell after subtracting the background intensity (Metamorph and ref. 23). To quantify the fluorescence of MART-1 at the Golgi area, the intensity of fluorescence at the Golgi ($I_g$; TGN46 as reference) and of the whole cell (total intensity $I_t$) was measured and the ratio $I_g/I_t$ was determined. All measures were performed using MetaMorph and averaged.

Quantification of immunogold labelling on ultrathin cryosections was performed by analysing the relative distribution of gold particles directly under the electron microscope in randomly selected micrographs and cells from at least three distinct grids[23]. For each condition, Gold particles labelling GFP-RAB6 in MNT-1 cells were counted and assigned to the compartment over which they were located. The definition of the distinct compartments was based on their morphology and their previous characterization by immunogold labelling with different organelle-markers. Melanosomes were counted based on morphology and pigment (stages-III/-IV) in randomly selected micrographs of RPE and assigned per 100 µm$^2$ square area.

The dynamics analyses of RAB6 vesicles (velocity, distance, pause duration, track time) were performed using ImageJ FIJI (plugin Manual Tracking). Tracks were defined as directional movement episodes (periods longer than two frames). Only tracks greater than 7 s and corresponding to an average of consecutive 8–10 frames were taken into consideration. Velocity was computed as displacement between frames divided by the interval between frames. Vesicles with velocity less than 0.2 µm s$^{-1}$ in consecutive frames were assigned with pause. Percentage in pause was defined as the total fraction of time of a vesicle in pause relative to its tracking time.

Quantification of protein levels on western blot was performed using FIJI software and intensities were normalized to loading control. To estimate MART-1 surface labelling, total surface cell intensities of randomly chosen cells were measured (MetaMorph) and values normalized with respect to control. Spinning disk confocal temporal acquisitions were processed using ICY—Edge preserving denoising[61], smoothing and manual background removal (ImageJ FIJI).

**Statistical analysis.** All data were generated from cells pooled from at least three independent experiments represented as (*n*), unless mentioned, in corresponding legends and represent experimental replicates. Statistical data were presented as means ± s.e.m. Multiple comparisons between experimental groups were analysed by ANOVA statistical test or Sidak's multiple comparisons test as mentioned, and statistical significance for two or three sets of data was determined by Student's *t*-test using GraphPad Prism, no sample was excluded. Animals and cells were randomly selected. Significant differences between control or treated samples in each experiment are indicated as *$P < 0.05$; **$P < 0.01$; ***$P < 0.001$; ****$P < 0.0001$. Only P value < 0.05 was considered as statistically significant.

**Data availability.** The authors declare that all relevant data supporting the findings of this study are available within the paper (and its Supplementary information file). Any raw data can be obtained from the corresponding author (C.D.) on reasonable request.

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

## Acknowledgements

We are grateful to Anna Akhmanova (Utrecht University, Utrecht, the Netherlands), Thierry Galli (Inst. Jacques Monod, Paris, France), Subba Rao Gangi Setty (Indian Institute of Science, Bangalore, India), Franck Perez and Ahmed El Marjou (Inst. Curie, Paris, France) and Michaël S. Marks (Children's Hospital of Philadelphia, Philadelphia, USA) for generous gifts of reagents; the Structure and Membrane Compartment lab for insightful discussions; Lucie Sengmanivong and Vincent Fraisier (CNRS, UMR 144, Inst. Curie) for image acquisition and deconvolution; Paolo Pierobon (INSERM, U932, Inst. Curie) and Xavier Heiligenstein (CNRS, UMR144, Inst. Curie) for live cell imaging processing and analyses; Matteo Gentili (INSERM U932, Inst. Curie) for shRNAs experiments. The authors greatly acknowledge the Nikon Imaging Center @ Institut Curie-CNRS and the PICT-IBiSA, member of the France-BioImaging national research infrastructure; and the Recombinant Antibody Platform of the Institut Curie (https://science.institut-curie.org/platforms/therapeutic-recombinant-antibodies/) for the production of human recombinant antibodies against RAB6:GTP (AA2), Giantin and anti-Human Cy3. This work has received support under the program 'Investissement d'Avenir' launched by the French Government and implemented by ANR with the references ANR-10-LBX-0038 and ANR-10-IDEX-0001-02 PSL, CEFIPRA Project (4903-1 to Subba Rao Gangi Setty and G.R.), Fondation pour la Recherche Médicale (FRM grant DEQ20140329491 Team label to G.R.), Ligue Nationale contre le Cancer and Labex CelTisPhyBio (to A.P.), CNRS, INSERM and Institut Curie.

## Authors contributions

All authors designed the experiments; A.P., S.M-L., S.B. did the experimental work; all authors analysed the experiments; A.P., G.R. and C.D. wrote the manuscript; all authors edited the manuscript.

## Additional information

**Competing interests:** The authors declare no competing financial interests.

