## [Peer Review File · Nature Communications]

Reviewers' Comments:

Reviewer #1 (Remarks to the Author)

The authors describe a new pathway for biosynthetic delivery of a subset of melanosomal proteins to melanosomes, that involves Rab6 at two different stages, vesicle exit from the Golgi and vesicle docking via ELKS, a rab effector previously involved in delivery of rab6 vesicles to the PM. The authors hypothesize that this is an example of the general secretory pathway being coopted by certain cells for transport to specific organelles, in this particular case for the synthesis of melanosomes by melanocytes. This is a very original proposal and the experiments are in general carefully done. However, various conceptual and experimental issues, including the statistical interpretation of some of the results reduce enthusiasm for this paper at this time.

1. The model-1. All experiments were performed in a melanoma cell line (MNT-1) and all conclusions may only apply to that particular cell line. The only observation in native tissue is decreased melanosome maturation in Rab6 KO mice (Fig2e, f), which may not necessarily be related to the proposed mechanism. (The authors use heterozygous as controls instead of wt animals, why?). To validate the model, the authors should at least show expression of endogenous Rab6 and ELKS in melanosomes in native tissue from WT mice. Furthermore, a functional experiment in Rab6 KO melanocytes in primary culture showing that known melanosome cargoes like the ones measured in Figure 5 do not get delivered to melanosomes would highly increase the confidence in the model being used and the results obtained with it.
2. The model-2. Exogenously expressed Rab6 was observed to colocalize with markers of melanosome in MNT-1 cells defining the contour of intracellular endosomes (Fig 1 b,c,d). However, the morphology of melanosomes looks very different in Fig 1a, where there was no transfection of melanosomal proteins and only endogenous proteins were studied. This suggests the possibility that expressing exogenous proteins has unintended effects on melanosome structure and perhaps function. There's no control examining this point. Moreover, please explain why in Fig 1h GFP-Rab6 protein is much less abundant than endogenous Rab 6 (relative to PNS) in the melanosome fraction. Furthermore, the claim that Rab6 mediates a biosynthetic pathway from TGN to melanosomes is based in part on a weak association with TfR with this pathway. In direct contrast with this postulate, Figure 1h shows a strong presence of TfR in the melanosome fraction. Finally, why do the authors see Rab6 in stage IV melanosomes in the EM image in Fig 1e, but not in the isolated melanosomes in Fig 1g? Is this picture representative? These concerns must be addressed as they question the validity of the chosen model to study the proposed mechanism.
3. The experiments in Figure 3 intend to prove that the Rab6-positive endosomes fuse with VAMP7 positive melanosomes. However, the evidence presented here does not fully support this conclusion. The sequence in Fig3c does not identify the ring-like structure as a melanosome, and none of the subsequent panels locate NPY-Venus in a Rab6 positive compartment. Actually, most Rab6 and NPY-Venus do not colocalize (Suppl. Fig 3c) and most VAMP7 does not colocalize with Rab6 (Fig1, Suppl fig3a). Hence, the identity of the compartment from where NPY-Venus is transferred to melanosomes as a Rab6 positive compartment is questionable.
4. The conclusion for the experiments shown in Fig 4 is that Rab6 and ELKS act as in same pathway to promote melanosome maturation. Most of this claim is based on the lack of additive effect in the double silencing in Fig 4b. However it is not clear how melanin content was measured, how it was quantified, and how it was normalized. I am concerned about whether a sample size with n=4 is enough to claim that the small decrease in the double silencing is not an additive effect. Indeed, a more appropriate experiment would be to disrupt the Rab6-ELKS interaction and to determine whether this negatively impacts melanosome maturation. The data only supports a role of ELKS in melanosome maturation but not necessarily that ELKS and Rab6 are in the same pathway. There seems to be a decrease in Rab6 presence in VAMP7 melanosomes (Fig 4h), but a possible increase in VAMP7 would also explain this and it was not measured. The dynamics of Rab6 vesicles changes after silencing ELKS (Fig 4k), but number of other reasons unrelated to the proposed mechanism might explain this. Overall, the evidence presented to conclude that Rab6 and ELKS act in the same pathway is not sufficient.
5. Figure 5 shows that known melanosome cargoes accumulate in the Golgi when Rab6 is silenced. This conclusion is based only on quantification of fluorescence intensity in the Golgi. It would be important, however, to obtain independent evidence for this point using a more quantitative biochemical method.
6. The model proposed raised some general question. A "philosophical" problem is that the general secretory mechanism proposed for delivery of melanosomal proteins would also bring many non

melanosomal proteins to the organelle. How would melanosomes control their identity when so many "contaminating" PM proteins are being delivered to the melanosome together with the "real" melanosome proteins. What is the advantage of having such a non specific mechanism contributing to the biogenesis of the melanosomes ?

7. The methods section is somewhat superficial and would not allow reproduction of the results (e.g culture conditions, antibody sources and dilutions, etc). A more complete methods section should be included, if space is short, as supplementary information.

Reviewer #2 (Remarks to the Author)

This paper provides new insight into the trafficking pathway responsible for melanosome biogenesis by showing the involvement of Rab6, a Golgi- and vesicle-localized GTPase and its binding partner ELKS in this process. Using a combination of fluorescence imaging, including live cell imaging, and electron microscopy, the authors show that Golgi-derived Rab6-positive vesicles transport specific cargoes to melanosomes. ELKS, an adaptor protein previously shown to be involved in docking of Rab6 vesicles with the plasma membrane, promotes Rab6 vesicle docking with the melanosomes. The importance of Rab6 for melanosome biogenesis is supported by the demonstration of a dramatic decrease in the number of pigmented melanosomes in retinal pigment epithelium of mice in which Rab6 was knocked out in the melanocyte lineage. If the authors can fully support their conclusions by data, this paper will certainly make a very important and exciting addition to the trafficking field.

Major comments:

1. The authors demonstrate that endogenous Rab6 localizes to the Golgi complex and cytoplasmic vesicles, while overexpressed GFP-Rab6 is also present on melanosomes. Since overexpressed Rabs often mislocalize, does this mean that finding GFP-Rab6 on melanosomes represents an overexpression artefact? Furthermore, quantification of colocalization is lacking.
2. The key conclusion of the paper is that Rab6 vesicles fuse with melanosomes, but the data are not entirely convincing. There is not a single example shown where a Rab6 vesicle actually fuses with a melanosome. Better two-color examples (preferably more than one) should be provided. Note that Figure 3c and movie S5 (switch from vesicle to ring-like structure) does not look convincing, possibly because the frame rate is not sufficiently high (acquisition of ~ 10 frames per second might be needed to observe vesicle fusion events). The data with the NPY marker (Fig. 3h) also do not look convincing as the fusion events are difficult to observe. It is also not clear whether NPY is indeed a good marker for Rab6 vesicles in melanocytes; the percentage of NPY vesicles colocalizing with Rab6, and vice versa, should be quantified. For proving the proposed model, high-quality data on Rab6 vesicle fusion with melanosomes are essential, because the loss of Rab6 might affect melanosome biogenesis in different ways, for example by regulating protein processing at the Golgi.
3. The localization of ELKS in melanocytes should be clarified. On p. 8 line 205, the authors first state that "ELKS localized to punctate structures nearby plasma membrane or pigmented melanosomes (Figure 4e)", and later (line 215) that "ELKS cooperates with RAB6 during melanosome biogenesis by associating with post-Golgi derived vesicles "en route" to and fusing with melanosomes". Previous work showed that ELKS is mainly present in plasma membrane-associated structures (Grigoriev et al Dev Cell 2007, Astro et al, J Cell Sci. 2014). Time lapse imaging data should be provided to convincingly demonstrate the presence of ELKS either on motile Rab6 vesicles or on melanosomes, to establish unambiguously that ELKS is indeed present on one of these membrane populations and can participate in their fusion.
4. The quantification of the effect of ELKS on the fusion of Rab6 vesicles with melanosomes does not seem to have been done in an appropriate manner. The authors have quantified the number of cells in which GFP-Rab6 colocalized with VAMP7-positive melanosomes and found a two-fold reduction, which they propose to fit with a 50% depletion of ELKS. This type of reasoning could only work if ELKS depletion would be an all-or-none effect, which is very unlikely to be the case. First, images of ELKS staining in ELKS-depleted cells side-by-side with controls should be provided. Second, percentage of Rab6 vesicles colocalizing with melanosomes and vice versa, in control and ELKS-depleted cells should be quantified. The total number of Rab6 vesicles should also be quantified, since it is expected to increase if ELKS indeed induces a fusion defect. Further, the reduction of Rab6 vesicle pausing in ELKS depleted cells might be due to a defect in fusion with the plasma membrane, and not with melanosomes, and thus does not provide additional support for the proposed trafficking route.
5. Model in Fig 5 - can the authors exclude that before fusion, ELKS is present on melanosomes and not on secretory carriers?

Minor comment: although not essential, the authors might consider making a CRISPR/Cas9 knockout of ELKS in MNT-1 cells, since it is likely to be easier and more efficient than protein depletion.

Reviewer #3 (Remarks to the Author)

Melanosomes are lysosome-related organelles specialized in the synthesis and storage of melanin pigment. How these organelles are put together is a subject of significant current interest. In this manuscript, the authors describe a novel route for transport of a subset of melanosomal proteins directly from the TGN to melanosomes, in addition to the better known route involving transport from endosomes to melanosomes. This direct route depends on the small GTPase Rab6 and its effector ELKS, previously implicated in secretory transport from the TGN to the plasma membrane. This study thus illustrates how a secretory route is adapted to deliver proteins to lysosome-related organelles in a specialized cell type.

The data are generally of high quality. However, the authors need to address (editorially or experimentally) the following issues:

Fig. 1: The IFM of RAB6 in comparison to melanosomal markers is not very convincing. Can the authors quantify colocalization or "apposition" over the entire cell? The insets in b-d show colocalization in a swollen vesicle that does not seem representative of a typical Rab6 vesicle. Would it be possible to show colocalization of a set of vesicles instead?

Fig. 1g: Although I understand that it is difficult to detect endogenous Rab6 by IEM, have the authors tried to detect it using purified melanosomes as in Fig. 1g? Perhaps the use of purified melanosomes affords greater sensitivity and lower background.

Fig. 2e: Did the authors see an increase in the number or proportion of immature melanosomes in RPE as they did in MNT-1 cells?

Fig. 3b: Between 7.1 and 7.9 s, did the Rab6 vesicle approach the VAMP7 vesicle and fused with it, or it just lost its Rab6?

Fig. 4a: Did siRNA to RAB6 also decrease ELKS? Is the difference reproducible and significant?

Fig. 4b: The authors conclude that the double RAB6-ELKS depletion is not significantly different from the single depletions, but it's not clear from the figure or the legends what statistical analysis was used to reach this conclusion.

Fig. 4g: Show siRNA control for this experiment.

Fig. 5a: Why do siRAB6 cells have increased TYRP2 and MART-1 levels?

Fig. S5b: It's probably not necessary to show these data since they concern a chimeric protein that is not used in further experiments.

Reviewer #4 (Remarks to the Author)

The paper by Patwardhan et al entitled "Routing of the Rab6 secretory pathway towards lysosome related organelles" investigates the possible function of Rab6 and ELKS in the biogenesis of melanosomes. The authors use a variety of approaches, present numerous results and make some interesting observations. Overall the data suggest a function for Rab6 and ELKS in the biogenesis of melanosomes. However, the data presented does not satisfactorily support the claim of a direct route from the TGN to melanosomes that is dependent on Rab6/ELKS and used by Tyrp2 and MART-1 but not by Tyr and Tyrp1.

Major points:

(1) The localization of Rab6 to melanosomes is not adequately supported by the data.

First, the localization of overexpressed GFP-Rab6 differs from that of endogenous Rab6 detected by immunofluorescence microscopy. Endogenous Rab6 does not show a "doughnut" type staining around the melanosome as observed for GFP-Rab6 suggesting GFP-Rab6 overexpression produces

an artifact.

Second, Figure 1h (top part) is problematic. The experiment uses cells transfected with GFP-Rab6, which are then lysed, subjected to melanosome enrichment, IP with an anti-GFP antibody, and WB analysis with various antibodies. If GFP-Rab6 localizes to melanosomes, the "IP" lane should show Tyrp1 but it does not. This goes directly against the idea that GFP-Rab6 is present in melanosomes and all the conclusions made from fluorescence and electron microscopy data involving GFP-Rab6. Likewise, if GFP-Rab6 and endogenous Rab6 localize to the same compartment, the "IP" lane should show endogenous Rab6 but this is not the case. It appears that GFP-Rab6 does not appropriately represent or localize as endogenous Rab6. This is a significant problem.

Third, these cells are overexpressing GFP-Rab6, therefore it is not clear if the ~23KDa band observed is truly endogenous Rab6 or the product of GFP-Rab6 cleaved during the preparation of cell lysates. Additionally, the presence of the Rab6 band in the "Mel input" lane is a weak evidence that the protein is indeed in melanosomes because the enrichment process is rather crude and this preparation likely includes other membrane bound compartments.

These concerns could be partially alleviated if immuno EM for endogenous Rab6 rather than overexpressed GFP-Rab6 were to show the protein localizes to pigmented melanosomes. This immuno EM analysis should include quantification and negative controls, such as melanocytes from the KO mouse or siRNA treated MNT-1 cells. The biochemical experiment should also be performed with control, non-transfected cells, and both immunoprecipitation and WB analysis should use antibodies to endogenous proteins (Rab6, Tyrp1, etc). These experiments could support the premise that endogenous Rab6 localizes to pigmented melanosomes. Still the question would remain regarding the validity of using overexpressed GFP-Rab6.

(2) The reduced pigmentation and biogenesis of pigmented melanosomes in Rab6 (and ELKS) deficient cells is convincing although no picture of the Rab6 KO mice was shown in order to observe if there is a fur or eye pigmentation defect (this should be provided). However, this result does not discriminate between a defect in transport specifically to melanosomes from an indirect effect of disrupting general secretory transport. Also, Rab6 siRNA cells show many stage I but no stage II melanosomes can be observed in the figure. This is also the case for ELKS deficient cells. Is this true? Stage I and II melanosomes should be further discriminated in the quantification. This is important because it appears that the problem is in the transition between stage I and II melanosomes rather than between stage II and III as proposed by the authors.

Pmel17 should be analyzed to better establish if there is a problem in generating stage II melanosomes. The HMB45 antibody recognizes the Pmel17 fragments present in the fibrils of stage II melanomes and should be compared in WB analysis and immuno EM of control, Rab6 and ELKS deficient cells. This experiment may help discriminate between a specific transport defect to stage II melanosomes (as proposed by the authors) from a non-specific defect in general secretion. Pmel17 does traffic, at least in part, through the plasma membrane "en route" to melanosomes and could very well be affected by the Rab6/ELKS deficiency. The authors should explore the possibility that Pmel17 transport and processing is defective.

(3) Do GFP-Rab6 labeled vesicles observed in Fig 3b actually fuse with the Vamp7 compartments? If so, what proportion of them does fuse?

The data in Figure 3c shows a GFP-Rab6 labeled vesicle apparently fusing with a compartment, which is not labeled but presumably would be a melanosome. This organelle should be identified as a melanosome by simultaneously imaging the pigment or a melanosome marker such as the one used in Fig 3b, Vamp7. It is hard to see whether there is a fusion event or the compartments are getting in/out of the focal plane. Also, the structure indicated as a vesicle in Fig 3c seems too big for a vesicle. A reference size bar should be included for comparison.

(4) Since many NPY-Venus vesicles are not baled by Rab6 (Fig S3c), it is unclear if the NPY-Venus delivered to melanosomes (Fig. 3f-h) was actually contained in Rab6 labeled vesicles or if Rab6 was needed for such transport.

(5) Biochemical studies with GFP-ELKS (Fig S4C) have a similar problem as described above for GFP-Rab6. The "IP" lane shows enrichment of GFP-ELKS relative to the "Mel Input" lane, however little Tyrp1 and Tyrp2 is present in the "IP" lane compared to the "Mel Input" and "Unbound" lanes. This result indicates the bulk of the overexpressed GFP-ELKS is not present in pigmented melanosomes. Consequently, it appears that organelles visualized in microscopy experiments with GFP-ELKS (Fig 4e-g and S4a,b,d) for the most part are not melanosomes. This issue is

exacerbated by the lack of quantification of these microscopy experiments. As discussed above for Rab6, this is a significant problem because it goes against the microscopy data obtained with the same GFP-ELKS construct.

(6) It is interesting that Tyrp2 and MART-1 (but not Tyr and Tyrp1) steady state levels are increased in Rab6 depleted cells. The authors interpret this result as suggesting Tyrp2 and MART-1 use a different pathway to target melanosomes and claim that such pathway is a novel, direct, Rab6-dependent route from the TGN to melanosomes. The data simply does not support such claims. Again this could be an indirect effect of generally altering normal exit from the Golgi apparatus. For instance, the data does not rule out traffic of Tyrp2 and MART through early endosomes. These claims are therefore an overstatement that should be eliminated or corrected, including in the abstract. Additionally, quantification of microscopy localization data should be presented (Fig 5c-d).

(7) Related to point (6). If Rab6 deficient cells have significantly less stage III/IV melanosomes and the steady state levels of Tyr and Tyrp1 are not affected, where are Tyr and Tyrp1 localized in these cells? It is quite possible that they are retained in the Golgi apparatus due to a general secretion defect. If this were true it would undermine the idea of a separate pathway to melanosomes and would also explain the pigmentation phenotype observed here. The authors need to compare in a quantitative fashion the localization of Tyr/Tyrp1 in control and Rab6/ELKS deficient cells to establish if their transport is affected or not.

(8) It is conceptually difficult to understand how two proteins that are ubiquitously expressed (Rab6 and ELKS) mediate transport to a tissue specific organelle (melanosome). How is the specificity achieved?

Minor points:

Title: this work explores melanosomes rather than multiple lysosome related organelles, therefore "lysosome related organelles" should be replaced with "melanosomes".

Abstract: "Our data together reveal for the first time that the secretory pathway can be directed towards intracellular organelles to ensure their biogenesis and function". Most secretory granules, which are intracellular organelles, derive from the secretory pathway. What is the novelty here?

In multiple occasions the text refers to Rab6 when in reality the experiment was performed with overexpressed GFP-Rab6. Given the concerns mentioned above this has to be clearly stated.

Figure 4g, the cell seems to have very low levels of GFP-Rab6. Is this a general issue in siELKS cells or just this example?

Figure 5g, the data is too preliminary to adequately support the model presented. The model should be eliminated.

Reviewers' comments:

Reviewer #1 (Remarks to the Author):

The authors describe a new pathway for biosynthetic delivery of a subset of melanosomal proteins to melanosomes, that involves Rab6 at two different stages, vesicle exit from the Golgi and vesicle docking via ELKS, a rab effector previously involved in delivery of rab6 vesicles to the PM. The authors hypothesize that this is an example of the general secretory pathway being coopted by certain cells for transport to specific organelles, in this particular case for the synthesis of melanosomes by melanocytes. This is a very original proposal and the experiments are in general carefully done. However, various conceptual and experimental issues, including the statistical interpretation of some of the results reduce enthusiasm for this paper at this time.

We would like to thank the reviewer for finding our work original and carefully done.

1. The model-1. All experiments were performed in a melanoma cell line (MNT-1) and all conclusions may only apply to that particular cell line. The only observation in native tissue is decreased melanosome maturation in Rab6 KO mice (Fig2e, f), which may not necessarily be related to the proposed mechanism.

The reviewer raised an important point. Hence, we have extensively reinforced our original study by performing several new experiments (IFM, EM, biochemistry) in primary human and mouse melanocytes (NHEM and NMM).

-First, using immuno-fluorescence microscopy (IFM) on both types of primary cells, we recapitulated our original findings showing that endogenous RAB6 associated to Golgi apparatus, intracellular vesicles as well as discrete spots that co-distributed with melanosomes (either melanin or melanosomal markers; **Revised Figures 1, S1-2**). Note that those RAB6 structures were also observed by live cell imaging upon expression in MNT-1 cells of an intra-body that specifically recognized activated RAB6:GTP (Nizak *et al*, *Science*, 2003) (**Revised Figure S1b** and **Movie S1**). In addition, we have performed a detailed quantification of the number of endogenous RAB6-positive spots associated to melanosomes and we have showed, consistently, that one melanosome associated in average with almost 1.5 RAB6 spots in primary melanocytes (mouse or human) and cell lines (MNT-1) (**pages 5 and 17**). As a control and as expected, the number of spots dropped in RAB6 KO melanocytes (**Revised Figures S2g-i, page 6**).

-Second, using immuno-electron microscopy (IEM) on melanosome-enriched fractions, our new data showed that endogenous RAB6 associated to the limiting membranes of melanosomes (both Stages III and IV, as quantified in **Revised Figure 1h**) in primary human melanocytes and the MNT-1 cell line (**pages 5-6**).

-Third, we replaced the previous Figure 1g (performed using Co-IP approach) to a direct detection of endogenous proteins in PNS (post-nuclear supernatant) and melanosome-enriched lysates (Mel). Using this biochemical approach, we demonstrated the association of endogenous RAB6 with melanosomes fractions (additionally revealed by IEM) also positive for melanosomal markers (TYRP1, TYRP2/DCT and MART-1). Those fractions were devoid of resident Golgi proteins (GM130) and cytosolic contaminants (GAPDH), showing that endogenous RAB6 did

not correspond to co-fractionation of Golgi membranes or cytosol and confirming its specific association with melanosomes (**Revised Figure 1j, page 6**).

-Fourth, decreasing the expression levels of RAB6, either in primary human melanocytes (siRNAs) or in primary melanocytes derived from RAB6 KO inducible mice (4-hydroxy-Tamoxifen-inducible mice; *Bardin et al 2015, Biol Cell*), recapitulated our previous results generated in MNT-1 cell lines. These melanocytes were isolated, treated with 4 hydroxy-tamoxifen to deplete RAB6 expression (5 days) and collected for experiments. Endogenous RAB6 expression is efficiently extinguished as revealed by IFM and WB (**Revised Figures S2g-j**) and associated with a dramatic drop in the number of pigmented stages (III and IV) relative to non-treated mice (**Revised Figures 2e-f and S2k**). In addition, those RAB6 KO melanocytes harbored more numerous immature and unpigmented melanosomes (stages-II). Conventional EM on primary human melanocytes siRNA-depleted for RAB6 recapitulated these data and showed a decrease in intracellular melanin content and in the number of pigmented melanosomes (**Revised Figures S2d-f**). See revised manuscript **page 6**.

-Fifth and to illustrate the role of RAB6 in delivering specific melanosomal cargoes from Golgi to melanosomes, we have confirmed the previous data generated in MNT-1 cells by showing the specific increased in protein level of the MART-1 and TYRP2 melanosomal cargoes in RAB6-depleted primary human melanocytes and the particular accumulation of MART-1 at the Golgi area relative to controls (**Revised Figures 6 and S5, pages 10-11**). Consequently using biochemistry, RAB6-depleted cells presented a decreased amount of MART-1 and TYRP2 in melanosome-enriched fractions relative to controls (**Revised Figure 6c**), supporting a default in Golgi to melanosomes trafficking of these two cargoes.

All together, these new data, now included in the revised manuscript, reinforce our previous observations and generalize our conclusions to normal human and mouse melanocytes.

The authors use heterozygous as controls instead of wt animals, why ?

We have carefully verified that we never observed phenotypical differences between the WT and Heterozygote (HET) animals (regarding for instance the pigmentation of the coat). Thus we have chosen to compare the homozygote (RAB6 KO) with littermate HET as controls.

To validate the model, the authors should at least show expression of endogenous Rab6 and ELKS in melanosomes in native tissue from WT mice.

RAB6 and ELKS expressions are known to be ubiquitous, and at least human and mouse melanocytes as well as melanocytic cell line expressed them (our study). However, performing immuno-histology on tissue sections labeled for endogenous RAB6 or ELKS and assigning their localization regarding melanosomes is extremely challenging. Indeed, apart from the limitation of available antibodies (especially for ELKS), the required resolution to define RAB6 and/ or ELKS fluorescent spots associated to melanosomes would be likely not sufficient.

We have thus decided to investigate this question by other approaches. In addition to the original data generated in MNT-1 and conditional RAB6 KO mice, we now add

data in primary human melanocytes (depleted or not using siRNAs) and primary RAB6 KO mice melanocytes (derived from inducible mice treated or not with 4-hydroxy-tamoxifen). Using different approaches (IFM, biochemistry and IEM), we showed that endogenous or overexpressed RAB6/ ELKS associated with melanosomes (**Revised Figures 1, 4**). Altogether the data generated (see also above the first point) appears to us extremely coherent, and thus strengthens and validates the original model.

Furthermore, a functional experiment in Rab6 KO melanocytes in primary culture showing that known melanosome cargoes like the ones measured in Figure 5 do not get delivered to melanosomes would highly increase the confidence in the model being used and the results obtained with it.

The reviewer is right and additional data have now been generated in primary human melanocytes. We have performed IFM on primary RAB6 KO mice melanocytes using two different MART-1 antibodies, but unfortunately those antibodies were unspecific in mice precluding our analysis.

Thus we used primary human melanocytes depleted for RAB6 (using specific siRNAs) and showed similarly a statistical increase of MART-1 localization at the Golgi area (as shown in MNT-1) (**Revised Figure 6f-g** and **S5e**); note that no TYRP2/DCT antibody is available to similarly examine its localization. Also by immunoblotting, RAB6-depleted primary human melanocytes increased specifically MART-1 and TYRP2 expression (**Revised Figure S5a**), while decreasing their association with melanosomes-enriched fractions (**Revised Figure 6c**), indicating a default in transport/ delivery of MART-1/ TYRP2 from Golgi to melanosomes (**page 11**).

2. The model-2. Exogenously expressed Rab6 was observed to colocalize with markers of melanosome in MNT-1 cells defining the contour of intracellular endosomes (Fig 1 b,c,d). However, the morphology of melanosomes looks very different in Fig 1a, where there was no transfection of melanosomal proteins and only endogenous proteins were studied. This suggests the possibility that expressing exogenous proteins has unintended effects on melanosome structure and perhaps function. There's no control examining this point.

First and to avoid possible confusion, we would like to stress that endogenous RAB6, RAB6 intra-body or GFP/mCherry-RAB6 associated with Golgi apparatus, derived vesicles and melanosomes in melanocytes but not endosomes.

Second, the morphology of melanosomes examined at the ultrastructural level by our diverse EM experiments on melanocytes (cell lines or primary) expressing or not exogenous RAB6 or on isolated melanosomes are not showing any alterations of the structure and pigmentation of melanosomes. For instance, the **Revised Figure 1** showed the GFP labeling on cells expressing GFP-RAB6. While GFP-RAB6 localized to the limiting membranes of melanosomes, their shapes and overall structures/ morphologies were similar to non-transfected cells or to other similar images published in many different studies. Also, we did not observe by WB change in the amount of key melanosomal proteins on melanosomal fractions (**Revised Figure 1**). Altogether we hope that this note will clarify the point raised by the reviewer

regarding the *'possibility of unintended effects by exogenous RAB6 on melanosomes structure and perhaps functions'*.

However, we believe that the reviewer refers here to the differences observed by IFM between the localization of endogenous RAB6 and RAB6:GTP (intra-body) (punctate structures associated to melanosomes; revealed by dark pigment seen by bright-field or melanosomal markers), and the typical 'donut'-like melanosome staining of exogenous (GFP/mCherry-RAB6) that overlapped almost completely with melanosomal markers. While these observations revealed a difference in the abundance of RAB6 (endogenous vs. exogenous) to melanosomes, they demonstrated that both endogenous and exogenous RAB6 associated with melanosomes. These results were now reinforced by showing similar observations in primary human and mouse melanocytes using different experimental setups (IFM, biochemistry and EM on cell sections or isolated melanosomes). Note that activated RAB6:GTP can be also observed as a donut-like structure co-localizing with melanosomes (**Revised Figure S1b** and **Movie S1**). In addition swapping the GFP tag for mCherry did not change the melanosomal localization of exogenous RAB6. Also we show that the melanosome association of GFP-RAB6 depends on its active state, because GTP-locked GFP-RAB6 version localized to melanosomes but not the GDP-locked form (**Revised Figures S1e-f, pages 5-6**). Finally and importantly, melanocytic cell lines expressing GFP-RAB6 but depleted for ELKS (**Revised Figure 5d, page 10**) showed a decreased colocalization between GFP-RAB6 and mCherry-VAMP7 (melanosomes), illustrating that the localization of the overexpressed RAB6 did not represent fortuitous localization but rather a specific one.

All together, we hope that our explanations and additional experiments are now reinforcing our data and addresses the concern raised by the reviewer.

Moreover, please explain why in Fig 1h GFP-Rab6 protein is much less abundant than endogenous Rab 6 (relative to PNS) in the melanosome fraction.

We agree with the reviewer that this observation can be misleading. The original figure 1h showed WB of differently exposed membranes. Indeed, the GFP-RAB6 and endogenous RAB6 membranes were revealed at different time exposures to avoid saturation of the GFP-RAB6 bands (due to over-exposure) or on the contrary barely detectable endogenous RAB6 (if membrane under-exposed).

Now, we replaced the original Figure 1h by similar experiment, run and revealed in parallel, showing the biochemical association of endogenous or exogenous RAB6 to melanosomes from primary human cells or MNT-1 cell line (**Revised Figure 1j, page 6**). These data showed that a fraction of endogenous and exogenous RAB6 associated with melanosomes (compare PNS vs. melanosome intensities of bands). Relatively to their respective PNS, GFP-RAB6 was indeed less enriched in melanosome fraction as compared to endogenous RAB6. However upon overexpression, GFP-RAB6 is also much more associated to Golgi membranes and cytosol that consequently lowers the melanosome/PNS ratio of association. Indirectly, this indicates that the association of GFP-RAB6 to melanosomes was not linear relative to its overall concentration reinforcing the specificity of its association.

Furthermore, the claim that Rab6 mediates a biosynthetic pathway from TGN to

melanosomes is based in part on a weak association with TfR with this pathway. In direct contrast with this postulate, Figure 1h shows a strong presence of TfR in the melanosome fraction.

We regret if we were not enough clear in our submitted manuscript. We do not claim that RAB6 mediates a TGN-melanosome pathway from the blots showed in Figure 1h. However, our conclusions raised from these blots were that a fraction of RAB6 associated with melanosomes.

Our model proposing a direct TGN-melanosome pathway is rather based on the collective interpretation of RAB6 association with the limiting membrane of melanosomes (IFM, EM, biochemistry), the dynamic association of Golgi-derived RAB6 vesicles with melanosomes (their short immobilization close to melanosomes and partial fusion), as well as known secretory components that either contribute to this pathway (ELKS) or follow that route (NPY). In addition to the accumulation of melanosomal cargoes (MART-1) in the Golgi, their decrease in melanosomes (MART-1 and TYRP2) in absence of RAB6, and their recognition as putative 'secretory cargoes' in non-pigment cells (HeLa cells lacking melanosomes), we believe that all these data support our model.

The relative association of TfR in the melanosome-enriched fraction (Figure 1h) was expected because previous studies had shown that Tf-positive recycling endosomes established direct contacts with melanosomes (*Delevoeye et al, 2009, J Cell Biol; Dennis et al, 2015, J Cell Biol*). Therefore TfR-positive endosomes were not contaminants in our preparation but rather represented the fractions of recycling endosomes that established tight contacts with melanosomes.

Altogether, we do not agree that this particular observation is in contrast with our proposed model, it just reflects the two pathways contributing to the maturation of melanosomes (the endosome- and Golgi-derived pathways). However and as mentioned above, this panel has now been replaced in the new **revised Figure 1j**, in which we have removed the TfR panels to avoid any possible confusion (**page 6**).

Finally, why do the authors see Rab6 in stage IV melanosomes in the EM image in Fig 1e, but not in the isolated melanosomes in Fig 1g? Is this picture representative? These concerns must be addressed as they question the validity of the chosen model to study the proposed mechanism.

The reviewer is correct in his/her observations and the point raised is due to a lack of clarity of our figures. The two figures together (1g+1e) were representative and reflected the association of RAB6 to pigmented stages-III and –IV melanosomes. We have now quantified the relative number of gold particles (labeling GFP-RAB6) associated to stage-III or –IV melanosomes on cells or when isolated (**Revised Figure 1g and i, page 6**). This showed that RAB6 associated mainly to stage-III relative to stage-IV. We have also added magnified areas of GFP-immuno-labeled MNT-1 cryosections presenting the GFP-RAB6 localization to stage-III and –IV (even by inverting the labeling of secondary protein-A gold). This data, in association with the increase of stage-II and decrease of stage-III observed in RAB6-depleted cells (human and mouse), support a role for RAB6 in the stage-II to –III maturation.

3-The experiments in Figure 3 intend to prove that the Rab6-positive

endosomes fuse with VAMP7 positive melanosomes. However, the evidence presented here does not fully support this conclusion.

As mentioned above, we are not presenting evidences proposing that endogenous RAB6 or GFP/mCherry-RAB6 localized to endosomes, and we tried to emphasize this information in the revised manuscript. RAB6 in melanocytes (MNT-1 cell line or primary human or mouse cells) associated with Golgi apparatus and derived vesicles and, in addition, to melanosomes (as shown and quantified by IEM in **Revised Figures 1f-g**). In **Figure 3** and associated **movies**, live cell panels showed dynamic association of RAB6-positive Golgi-derived vesicles with RAB6- and VAMP7-positive melanosomes.

The sequence in Fig3c does not identify the ring-like structure as a melanosome,

The reviewer is absolutely right that we don't identify the ring-like structure as a melanosome. In fact, we have assumed it, due to its size and shape by optical microscopy. This is also why we were cautious in the submitted text by saying (page 7, lane 161-164) "*RAB6 fluorescence disappearing from the docked vesicles (arrows) increased concomitantly at ring-like structures (arrowheads) (Figure 3c and Movie S5), most likely indicative of fusion events occurring between RAB6 vesicles and melanosomes.*"

However we have realized that we were not enough clear about the frequency and different types of fusion events that were observed between RAB6 vesicles and melanosomes. Our careful examination of RAB6-positive dynamics relative to melanosomes indicate that they fuse via kiss&run mechanisms rather than a complete fusion and consumption of vesicles into melanosomes. We now show that RAB6 vesicles emerging from the Golgi directly contact a melanosome, stay for a while, and then occasionally meet another melanosome (**Movie S3, page 7**). Together, this indicates that RAB6 vesicles can likely undergo successive contacts and kiss&run events with distinct melanosomes. In the revised manuscript, we have removed this 'fusion panel' (submitted Figure 3c and movie S5) to first avoid confusion and second to reinforce the plausible suggestion that RAB6 vesicles undergo kiss&run events with melanosomes.

The kiss&run events are now illustrated by the measure of the NPY-Venus fluorescence intensity (normalized to the one of mCherry-VAMP7) that showed a raise of NPY intensity within a melanosome at the time where the NPY-vesicle established tight contact with the VAMP7-positive melanosome (**Revised Figures 3h-i and Movie S6**). This data demonstrates the partial transfer of NPY from the vesicle to the melanosome and thus a kiss&run event. In addition to the IEM showing NPY in the lumen of pigmented melanosomes and the **revised Movie S3**, our data indicate that one vesicle can likely undergo multiple and successive kiss&run events with different melanosomes (**page 8**).

Therefore, we now amended this point by slightly modifying the text in order to reveal that kiss&run behaviour is the most common observations.

and none of the subsequent panels locate NPY-Venus in a Rab6 positive compartment. Actually, most Rab6 and NPY-Venus do not colocalize (Suppl.

Fig 3c) and most VAMP7 does not colocalize with Rab6 (Fig1, Suppl fig3a). Hence, the identity of the compartment from where NPY-Venus is transferred to melanosomes as a Rab6 positive compartment is questionable.

We are now providing quantification for RAB6 co-localization with NPY in MNT-1 cells (Pearson correlation in **Revised Figures S3b-c, page 7**). As expected from previous studies in non-pigment cells showing similar RAB6/NPY co-localization in non-pigment cells (60-70% co-localization; *Grigoriev I, 2007, Dev Cell*), this data shows a significant fraction of the RAB6-Golgi derived vesicles that indeed associates with the secretory cargo NPY in melanocytes.

Studies from several groups (Marks, Di Pietro, Fukuda or Setty labs) have shown that VAMP7 associates mainly with melanosomes in melanocytes, and we show here that RAB6 associated with melanosomes by EM. We now provide two types of quantifications demonstrating the co-localization of RAB6 with melanosomes at the optical level. First almost 1.5 endogenous RAB6-positive structure co-associates per melanosome (**Revised Figures 1c, S2h**) in human and mouse primary melanocytes as well as in MNT-1 cell line. Note that Pearson coefficient was not applicable in that situation due to the discrete nature of the RAB6 vesicles as compared to the relative large size of the melanosomes. Second, exogenous RAB6 co-localized extensively with VAMP7 melanosomes (Pearson coefficient in **Revised Figure 1d, page 5**). Note also that this co-localization is dropped in ELKS-depleted cells (**Revised Figure 5d, page 10**) demonstrating the specificity of the RAB6 association to melanosomes, the dependency to ELKS and its specific co-localization with VAMP7.

4. The conclusion for the experiments shown in Fig 4 is that Rab6 and ELKS act as in same pathway to promote melanosome maturation. Most of this claim is based on the lack of additive effect in the double silencing in Fig 4b.

We do not fully agree with the reviewer. Several complementary data lead us to propose that RAB6 and ELKS cooperate for melanosome maturation. We have now added new data reinforcing our conclusions.

-First, live cell imaging showed that RAB6 and ELKS co-localized dynamically on vesicles emerging from the Golgi apparatus (**Revised Figures 4i, S4c and Movie S7, right panel, page 9**).

-Second, the number of ELKS-depleted melanocytes presenting RAB6 localization at melanosomes is decreased relative to control (**Revised Figure 5d**). We provide now co-localization (Pearson coefficient) between RAB6 and VAMP7 in control- and ELKS-depleted MNT-1 cells (**Revised Figure 5d, Page 10**) that indicate the need of ELKS to associate RAB6 to melanosomes.

-Third, we include data on cells co-expressing mCh-RAB6 together with GFP-ELKS- Δ Ct, a C-terminal deleted version of ELKS lacking its RAB6 binding domain (*Monier et al, Traffic, 2002*). We confirmed that GFP-ELKS- Δ Ct did not co-localize with RAB6 vesicles emerging from the Golgi as compared to WT form. Also MNT-1 cells expressing GFP-ELKS- Δ Ct did not depict RAB6 at donut-like melanosomes without observable impact on the generation of RAB6 vesicles (**Revised Figures 4i-j, and Movie S7, right panel, pages 9-10**), as already described (*Grigoriev et al, 2007, Dev Cell; Grigoriev et al, 2011, Cur Biol*). This data confirm the siRNA ELKS experiments and indicate that the interaction of ELKS with RAB6 is required for localizing RAB6 to

melanosomes.

-Fourth, our observations on cells depleted for RAB6, ELKS or both together, analyzed by conventional EM showed a similar drop in the number of pigmented melanosomes, as illustrated by the new quantification of melanosomal stages (**Revised Figures 4c-d**); note the increase of unpigmented stages-II, while pigmented ones decreased (**page 9**).

-Fifth, the lack of additive effect (melanin production, **Revised Figure 4b**) in the double KD conditions reinforces the other observations and supports our proposed model.

However it is not clear how melanin content was measured, how it was quantified, and how it was normalized. I am concerned about whether a sample size with n=4 is enough to claim that the small decrease in the double silencing is not an additive effect.

The reviewer is correct and we regret, but there was an issue with the materials and methods file. We believe that for some reason independent of our wish, it was not accessible to the reviewers. Indeed, the protocol used for melanin estimation was briefly mentioned within that supplementary file by referring to previous studies from our group (*Delevoye et al, J Cell Biol, 2009*). We modified the text in **page 4** of the revised supplementary methods document and repeat here our additional note.

“For melanin assay, cells were disrupted by sonication in 50 mM Tris-HCl, pH 7.4, 2 mM EDTA, 150 mM NaCl, 1 mM dithiothreitol, (with protease inhibitor cocktail, Roche). Pigment was pelleted at 20,000 g for 15 min at 4°C, rinsed once in ethanol/ether (1:1), and dissolved in 2M NaOH/20% dimethylsulfoxide at 60°C. Melanin content was measured as optical density at 490 nm (Spectramax 250, Molecular Devices).”

Regarding sample size of N=4, we have now repeated this particular experiment and the obtained result (N=6) reinforces our previous conclusions (**Revised Figure 4b, pages 8-9**), namely the double RAB6 and ELKS extinction do not synergize statistically the drop in melanin production relative to single KD condition, indicating together with previously described results that RAB6 and ELKS cooperate during melanosome maturation.

Indeed, a more appropriate experiment would be to disrupt the Rab6-ELKS interaction and to determine whether this negatively impacts melanosome maturation. The data only supports a role of ELKS in melanosome maturation but not necessarily that ELKS and Rab6 are in the same pathway.

One of the major issues resides in our capacity of transfecting efficiently melanocytes and in maintaining them enough time in culture (almost a week) to be able to monitor the impact on the melanosome maturation and pigmentation. The average efficiency of transfection for the expression of exogenous proteins is almost 10%. In addition to the fact that pigmented melanosomes are very long-lived, it makes difficult to assess the pigmentation process biochemically, and the biogenesis of melanosomes by EM. One alternative might be to generate ELKS KO melanocytes followed by generating stable melanocytes expressing ELKS mutants that do not interact with RAB6.

However, we can suspect long-term side effects or adaptations on the cell physiology by altering an important trafficking protein for such a long durations.

Thus, to address this question, we chose another strategy. We used the overexpression of the GFP-ELKS- Δ Ct that did not contain its RAB6 binding domain (Monier et al, Traffic, 2002). As mentioned above, the deleted form of ELKS did not co-distribute anymore with dynamic RAB6 vesicles (**Revised Movie S7**), but still associated to the very static plasma membrane structures (as already described in Grigoriev et al, Dev Cell, 2007). Accordingly to our model and to our siRNA ELKS experiments, mCh-RAB6 failed to associate with melanosomes in GFP-ELKS- Δ Ct-expressing cells (**Revised Figure 4j**), as indicated by the absence of observable donut-like structures; that was present in GFP-ELKS-WT expressing cells (**Revised Figure 4i**). This shows that ELKS requires its RAB6 binding domain to localize to RAB6 vesicles and to promote the association of RAB6 to melanosomes. Thus this indicates that the ELKS-RAB6 interaction is most likely needed for the docking and subsequent fusion of RAB6 vesicles with melanosomes. In addition to other data, this result supports the role of ELKS and RAB6 within the same pathway (**pages 9-10**).

There seems to be a decrease in Rab6 presence in VAMP7 melanosomes (Fig 4h), but a possible increase in VAMP7 would also explain this and it was not measured.

The reviewer is right, this was a possibility that we have addressed by testing the expression of VAMP7 in RAB6- and/ or ELKS-depleted melanocytes (**Revised Figure S4a, page 9**) as well as its presence at melanosomes in our overexpression studies. We did not observe any change in the expression of VAMP7.

The dynamics of Rab6 vesicles changes after silencing ELKS (Fig 4k), but number of other reasons unrelated to the proposed mechanism might explain this.

In order to address at our best the comment of the reviewer, we counted the relative number of RAB6 vesicles observed in control- or ELKS-depleted cells and did not observe differences (**Revised Figures 5a-b, page 10**).

Regarding the 'other reasons unrelated to the proposed mechanisms', we are not aware of any experimental approach that can address this comment. We provide a body of evidence, using different technical approaches (IFM, biochemistry, EM, live cell imaging) on cells depleted by specific siRNAs or KO, overexpressing WT or deleted versions, that all converge to the proposed model. Even if we cannot rule out any other unidentified reason to explain our phenotype, the most reasonable one is that RAB6 and ELKS cooperate in the melanosome biogenesis as already described at the plasma membrane along the secretory pathway.

Overall, the evidence presented to conclude that Rab6 and ELKS act in the same pathway is not sufficient.

We respectfully disagree with the reviewer and hope that our explanations, clarifications and additional data make stronger our proposed model.

5. Figure 5 shows that known melanosome cargoes accumulate in the Golgi when Rab6 is silenced. This conclusion is based only on quantification of fluorescence intensity in the Golgi. It would be important, however, to obtain independent evidence for this point using a more quantitative biochemical method.

As suggested by the reviewer, we show now by biochemical methods that knocking down RAB6 reduces the amounts of MART-1 and TYRP2/DCT in melanosomes-enriched fractions, while increasing them in the post-nuclear fractions (consisting in cytosol, Golgi membranes, intracellular vesicles and melanosomes) (**Revised Figure 6c**). In addition to our fluorescent data showing that MART-1 localized more in the Golgi upon RAB6 depletion (MNT-1 and primary human melanocytes), this new result reinforces our model proposing that RAB6 contributes to sorting/ exit of melanosomal cargoes out of Golgi and towards melanosomes (**page 11**).

6. The model proposed raised some general question. A "philosophical" problem is that the general secretory mechanism proposed for delivery of melanosomal proteins would also bring many non melanosomal proteins to the organelle. How would melanosomes control their identity when so many "contaminating" PM proteins are being delivered to the melanosome together with the "real" melanosome proteins. What is the advantage of having such a non specific mechanism contributing to the biogenesis of the melanosomes ?

This "philosophical" and extremely interesting question is the major challenge that highly specialized cells (including melanocytes but not only) have to face. How to ensure their own specificity, while using/ exploiting similar transport routes, same carriers and identical trafficking/ sorting machineries as other (perhaps) less specialized cells? For instance in neuron, how are specific receptors delivered to the axon or dendrites? Many studies point out that specific sorting of cargoes, their interactions with molecular motors or modifications of the cytoskeleton contribute to this specificity of trafficking. We believe, given our previous studies and those by others, that such general assumption might also apply to the melanocyte.

Regarding our study, we would like to comment on the similarities and differences we observed between the Golgi-melanosomes pathway and the current knowledge on the conventional secretory pathway (Golgi-plasma membrane).

Similarly to the secretory pathway, we did not observe a change in the number of RAB6 vesicles formed upon ELKS depletion (**Revised Figure 5b**), corroborating previous data (*Grigoriev et al, 2007, Dev Cell and 2011, Cur biol*). As in Grigoriev et al (2007), ELKS-ΔCt-expressing cells did not abolish the transport of RAB6 vesicles towards the periphery (**Revised Figure 4i**, note the accumulation of RAB6 fluorescence at the cell periphery). Also, depleting ELKS decreased the pause duration at melanosomes (**Table 1**), as shown at plasma membrane (*Grigoriev et 2007*).

However, some differences were highlighted by our study. (1) Compared with the findings by Grigoriev et al (*2007 and 2011*), ELKS associates with dynamic RAB6 vesicles emerging from the Golgi (**Revised Figure 4i, Movie S7**). (2) In contrast to secretory cargoes (i.e. NPY), RAB6 in melanocytes operates at a step impacting the

sorting/ exiting of melanosomal cargoes (i.e. MART-1) away from the Golgi (**Revised Figures 6f-h and S5e**). (3) While RAB6 depletion delays the secretion of cargoes at plasma membrane (more or less dramatically depending on the cargo that is monitored (*Grigoriev et al 2007 and 2011*)), it strongly impairs the delivery of melanosomal cargoes to melanosomes, as revealed by melanin decrease, maturation defect of melanosomes and decrease amount of MART-1 and TYRP2 to melanosomes (**Revised Figures 2, 4 and 6c**). (4) Similarly ELKS that is not required for the secretion per se (*Grigoriev et al 2011*), tightly controls in melanocytes (as RAB6 did) the transport and fusion of RAB6 vesicles with melanosomes (**Revised Figure 4**).

These results show that the Golgi-melanosomes pathway shares only some features with the “conventional” secretory pathway and thus represents an adaptation/ exploitation of it. The underlying mechanisms and machineries leading to this adaptation most likely represent the key to understand its specificity and are currently under investigation. Our results indicate that the specificity is achieved at the Golgi level (via RAB6 for sorting cargoes), at the transport step (via RAB6 and ELKS for contacting melanosomes) and at docking step (via ELKS for stabilizing the vesicle-melanosomes interaction).

By delivering selected melanosomal cargoes to this lysosome-related organelle, this novel pathway is thus a specific mechanism that does segregate between plasma membrane components to be secreted and melanosomal cargoes. Hence this ensures an advantage by promoting a tuning of melanosomes maturation without losing their identity by adding ‘contaminating’ components. And thus there is certainly a real advantage for the melanocytes to tune/ adapt this conventional pathway in a specific route. The NPY experiments are here a ‘trick’ that we have used to follow the fusion of RAB6 vesicles with melanosomes based on the fact that melanocytes do not express endogenous NPY.

We have added a note in the Discussion section and slightly modified the final model that summarizes the differences/ similarities with the secretory pathway to reinforce the idea of its exploitation (**Revised Figure 6h and Page 15**).

7. The methods section is somewhat superficial and would not allow reproduction of the results (e.g culture conditions, antibody sources and dilutions, etc). A more complete methods section should be included, if space is short, as supplementary information.

We agree and we regret because it seems, that independently of ours wish, the supplementary material and methods files were apparently not accessible online during our first submission. The methods associated to the main text corresponded to the image and statistical analyses; all the other methods were detailed in the supplementary document, but not accessible. We hope that this will not happen during resubmission of the revised version.

We would like to thank the reviewer for her/ his very detailed analysis of our work. We think that additional data are strengthening our proposed model and we hope that our answers will satisfy her/ him.

Reviewer #2 (Remarks to the Author):

This paper provides new insight into the trafficking pathway responsible for melanosome biogenesis by showing the involvement of Rab6, a Golgi- and vesicle-localized GTPase and its binding partner ELKS in this process. Using a combination of fluorescence imaging, including live cell imaging, and electron microscopy, the authors show that Golgi-derived Rab6-positive vesicles transport specific cargoes to melanosomes. ELKS, an adaptor protein previously shown to be involved in docking of Rab6 vesicles with the plasma membrane, promotes Rab6 vesicle docking with the melanosomes. The importance of Rab6 for melanosome biogenesis is supported by the demonstration of a dramatic decrease in the number of pigmented melanosomes in retinal pigment epithelium of mice in which Rab6 was knocked out in the melanocyte lineage. If the authors can fully support their conclusions by data, this paper will certainly make a very important and exciting addition to the trafficking field.

We thank the reviewer for his/ her supportive comment.

Major comments:

1. The authors demonstrate that endogenous Rab6 localizes to the Golgi complex and cytoplasmic vesicles, while overexpressed GFP-Rab6 is also present on melanosomes. Since overexpressed Rabs often mislocalize, does this mean that finding GFP-Rab6 on melanosomes represents an overexpression artefact? Furthermore, quantification of colocalization is lacking.

We reinforce now our original results showing that endogenous RAB6 and activated RAB6:GTP (intra-body), GFP/mCherry-RAB6-WT and GTP-locked form associated with melanosomes in melanocytes.

We believe that the reviewer refers here to the differences observed by IFM between the localization of endogenous RAB6 and the activated form (intra-body) (punctate structures associated to melanosomes; revealed by dark pigment seen by bright-field or melanosomal markers), and the typical 'donut'-like melanosome staining of exogenous (GFP/mCherry-RAB6) that overlapped almost completely with melanosomal markers. While these observations revealed a difference in the abundance of RAB6 (endogenous vs. exogenous) to melanosomes, they demonstrated that both endogenous and exogenous RAB6 associated with melanosomes.

These results were now reinforced by showing similar observations in primary human and mouse melanocytes using different experimental setups (IFM, biochemistry and EM on cell sections or isolated melanosomes, **Revised Figures 1 and S1**). Note that endogenous RAB6 localized to the limiting membranes of isolated melanosomes from primary human melanocytes and the MNT-1 cell line (**Revised Figure 1h**). By expressing a fluorescent intra-body recognizing endogenous activated RAB6:GTP (Nizak *et al*, *Science*, 2003), we showed by live cell imaging that endogenous RAB6:GTP localized to discrete structures associated with melanosomes as in fixed cells (**Movie S1**), that can be found also co-localizing as a donut with VAMP7 melanosomes (**Revised Figure S1b**). In addition swapping the GFP tag for mCherry did not change the melanosomal localization of exogenous RAB6 (**Revised Figures**

1, S1). Also we show that the melanosome association of GFP-RAB6 depends on its active state, because the GTP-locked GFP-RAB6 version localized to melanosomes but not the GDP-locked form (**Revised Figures S1e-f, pages 5-6**). Finally and importantly, melanocytic cell lines expressing GFP-RAB6 but depleted of ELKS (**Revised Figures 5a, 5d, page 10**) showed a decreased co-localization of GFP-RAB6 with mCh-VAMP7 (melanosomes), illustrating that the localization of the overexpressed RAB6 did not represent fortuitous localization but rather a specific one.

We now provide two independent quantifications demonstrating the co-localization of RAB6 with melanosomes at the optical level. First almost 1.5 endogenous RAB6-positive fluorescent structure co-associates per melanosome (**Revised Figures 1c, S2h**) in primary human and mouse melanocytes and human MNT-1 cell line. Note that Pearson coefficient was not applicable in that situation due to the discrete nature of the RAB6 vesicles as compared to the relative large size of the melanosomes. Second, exogenous RAB6 co-localized extensively with VAMP7 melanosomes as revealed by its Pearson coefficient (**Revised Figure 1d, page 5**). This co-localization is dropped in ELKS-depleted cells (**Revised Figure 5d, page 10**) demonstrating the specificity of the GFP-RAB6 association to melanosomes and its specific co-localization with VAMP7.

All together, we hope that our explanations and additional experiments are now reinforcing our data and address the point raised by the reviewer.

2. The key conclusion of the paper is that Rab6 vesicles fuse with melanosomes, but the data are not entirely convincing. There is not a single example shown where a Rab6 vesicle actually fuses with a melanosome. Better two-color examples (preferably more than one) should be provided. Note that Figure 3c and movie S5 (switch from vesicle to ring-like structure) does not look convincing, possibly because the frame rate is not sufficiently high (acquisition of ~ 10 frames per second might be needed to observe vesicle fusion events). The data with the NPY marker (Fig. 3h) also do not look convincing as the fusion events are difficult to observe.

The reviewer is absolutely right that we don't identify the ring-like structure as a melanosome. In fact, we have assumed it, due to its size and shape by optical microscopy. This is also why we were cautious in the submitted text by saying (page 7, line 161-164) "*RAB6 fluorescence disappearing from the docked vesicles (arrows) increased concomitantly at ring-like structures (arrowheads) (Figure 3c and Movie S5), most likely indicative of fusion events occurring between RAB6 vesicles and melanosomes.*"

However we have realized that we were not enough clear about the frequency and different types of fusion events that were observed between RAB6 vesicles and melanosomes. Our careful examination of RAB6-positive dynamics relative to melanosomes indicate that they fuse via kiss&run mechanisms rather than a complete fusion and consumption of vesicles into melanosomes. We now show that RAB6 vesicles emerging from the Golgi directly contact a melanosome, stay for a while, and then possibly meet another melanosome (**Movies S3, S5, page 7**). Together, this indicates that RAB6 vesicles can likely undergo successive contacts and kiss&run events with distinct melanosomes. In the revised manuscript, we have

removed this ‘fusion panel’ (submitted Figure 3c and movie S5) to first avoid confusion and second to reinforce the plausible suggestion that RAB6 vesicles undergo kiss&run events with melanosomes.

The kiss&run events are now illustrated by the measure of the NPY-Venus fluorescence intensity (normalized to the one of mCherry-VAMP7) that showed a raise of NPY intensity within a melanosome at the time where the NPY-vesicle established tight contact with the VAMP7-positive melanosome (**Revised Figure 3h-i** and **Movie S6**). This data demonstrates the partial transfer of NPY from the vesicle to the melanosome and thus a kiss&run event. In addition to the IEM showing NPY in the lumen of pigmented melanosomes (**Revised Figure 3e**) and the **Movie S3**, our data indicate that one vesicle can undergo multiple and successive kiss&run events with different melanosomes (**page 8**).

We used NPY instead of RAB6 to show the fusion of vesicles with melanosomes for two reasons. First, a kiss&run fusion process is partial, thus the exchange of membrane-associated protein is limited and rather difficult to observe relative to a soluble component (like NPY). The second reason is that RAB6 localizes to the vesicle but also to the melanosome (due to former rounds of fusion), thus it became extremely challenging to assess the exchange of RAB6 between two compartments that are already positive for RAB6, thus explaining why we have used NPY instead of RAB6 as reporter molecules.

Knowing that most of the NPY vesicles co-localized with RAB6 (new quantification provided in **Revised Figure S3b-c**), and that RAB6 dynamic contacts and associations with melanosomes are decreased upon ELKS depletion or the expression of its RAB6 binding mutant (**Revised Figures 5, 4i-j**, and **Movie S7**), we believe that we provide altogether a body of evidence indicating that RAB6 vesicles fuse with melanosomes through kiss&run.

Therefore, we now amended this point by slightly modifying the text in order to reveal that kiss&run behaviour is the most common observations (revised manuscript **pages 7-10**)

It is also not clear whether NPY is indeed a good marker for Rab6 vesicles in melanocytes; the percentage of NPY vesicles colocalizing with Rab6, and vice versa, should be quantified.

NPY is a well-established exocytic RAB6-dependent cargo as already published before (*Grigoriev I, 2007, Dev Cell*). In order to monitor the dynamic exchange of content from RAB6 vesicles to the melanosomes, we looked for a soluble reporter molecule packaged specifically into RAB6 vesicles. The pathway we described corresponds to a post-Golgi route that ends in melanosome. Until now, the main RAB6-dependent post-Golgi route described so far is the Golgi-Plasma Membrane secretory pathway. Given that NPY is a canonical cargo of this RAB6 secretory pathway, we used NPY-Venus as a soluble and fluorescent reporter molecules allowing us to track possible fusion events between post-Golgi RAB6 vesicles and melanosomes. Thanks to this “artificial system”, we were able to monitor a partial but consistent exchange of fluorescence from NPY-positive vesicles to melanosomes (**Revised Figure 3i** and **Movie S6, page 8**), as well as observing NPY-Venus staining within the lumen of pigmented melanosomes (**Revised Figure 3e**). This result led us to propose that RAB6 vesicles can fuse with melanosomes and that the

Golgi-Melanosome pathway might share properties of the secretory pathway or might correspond to its adaptation in a specialized cell type. In a second part of the manuscript, our hypothesis was reinforced by our demonstration of the implication of ELKS in that pathway.

RAB6 localizes abundantly to the Golgi apparatus and also to vesicles that follow the retrograde route (endosome to Golgi and Golgi to endoplasmic reticulum), thus we can only expect a partial overlap between RAB6 and NPY. We are now providing quantification for NPY co-localization with RAB6 in MNT-1 (Pearson correlation coefficient, **Revised Figures S3b-c, page 8**). As expected from previous studies showing similar RAB6/NPY co-localization in non-pigment cells (*Grigoriev I, 2007, Dev Cell*), this data shows a significant fraction of the RAB6-Golgi derived vesicles that indeed associates with the exogenously expressed secretory cargo NPY in melanocytes.

We now add a note (**pages 7-8**) explaining the rationale of using NPY as a tool to track the 'secretory pathway' or its adapted version in melanocytes.

For proving the proposed model, high-quality data on Rab6 vesicle fusion with melanosomes are essential, because the loss of Rab6 might affect melanosome biogenesis in different ways, for example by regulating protein processing at the Golgi.

As detailed above, we decided to use NPY, instead of RAB6, as a reporter molecule allowing us to track the fusion events. The new data showed rather a partial transfer of fluorescence than a complete consumption of the vesicle into melanosome, arguing a kiss&run mechanism (see above our answers to the same reviewer).

Additionally and given that RAB6 might control the Golgi homeostasis in melanocytes, we provide additional data regarding Golgi functions. A disruption of Golgi functions could lead to general defect impacting globally transmembrane protein glycosylation and/or trafficking.

By immunoblotting, we did not observe a change in the migration patterns of TYRP1, TYR or PMEL (three heavily glycosylated transmembrane melanosomal proteins), indicating no general defect on their glycosylation and thus no global malfunctions of the Golgi (**Revised Figures S5a-c, page 11**). In addition, we showed that PMEL is properly processed (between Golgi and endosomes) as illustrated by the similar amount of PMEL-N/-C (**Revised Figure 5b**) and of the HMB45-positive PMEL fibrils (see below). This indicates that the processing and fibrils formation in stage-II melanosomes are not impacted by the loss of RAB6 expression and illustrates that the overall Golgi function is not perturbed (glycosylation, sorting, trafficking or homeostasis).

3. The localization of ELKS in melanocytes should be clarified. On p. 8 line 205, the authors first state that "ELKS localized to punctate structures nearby plasma membrane or pigmented melanosomes (Figure 4e)", and later (line 215) that "ELKS cooperates with RAB6 during melanosome biogenesis by associating with post-Golgi derived vesicles "en route" to and fusing with melanosomes". Previous work showed that ELKS is mainly present in plasma membrane-associated structures (Grigoriev et al Dev Cell 2007, Astro et al, J Cell Sci. 2014). Time lapse imaging data should be provided to convincingly demonstrate the presence of ELKS either on motile Rab6 vesicles or on melanosomes, to establish unambiguously that ELKS is indeed present on one of these membrane populations and can participate in their fusion.

The reviewer is correct that we were not clear enough. In melanocytes, ELKS localizes to very static and stable plasma membrane structures as already described in the two mentioned papers, but also to RAB6-positive vesicles and melanosomes.

-By biochemistry, endogenous ELKS associated with the melanosome-enriched fractions (**Revised Figure 4h**) as also confirmed by IFM (**Revised Figure 4e**) or by IEM of GFP-ELKS expressing MNT-1 cells (**Revised Figures 4g**). See **page 9**.

-By live cell imaging, we show now that GFP-ELKS associated to dynamic RAB6 vesicles emerging from the Golgi area (**Revised Figure 4i**, and **Movie S7**). However GFP-ELKS was barely detectable at the Golgi per se. Knowing that ELKS is a RAB6 effector (*Monier, 2002, Traffic*), this indicates that ELKS might be recruited by RAB6 on post-Golgi-derived vesicles. Indeed GFP-ELKS- Δ Ct, GFP-ELKS lacking its RAB6 binding domain (described in *Monier et al, Traffic 2002 and Grigoriev et al, Dev Cell 2007*) did not co-distribute with RAB6 vesicles (**Revised Figure 4i** and **Movie S7**), whereas it associated to cortical patches at plasma membrane. This shows that the association of ELKS to dynamic post-Golgi vesicles requires the binding to RAB6, but not its recruitment to plasma membrane. Interestingly, cells expressing this truncated ELKS did not harbor RAB6 at melanosomes (**Revised Figure 4j**, thus acting like a dominant negative version of ELKS, **Movie S7**), recapitulating our first observation using ELKS siRNA-treated melanocytes.

On the revised manuscript, we add notes **pages 9-10** of the revised manuscript that recapitulate our different observations.

4. The quantification of the effect of ELKS on the fusion of Rab6 vesicles with melanosomes does not seem to have been done in an appropriate manner. The authors have quantified the number of cells in which GFP-Rab6 colocalized with VAMP7-positive melanosomes and found a two-fold reduction, which they propose to fit with a 50% depletion of ELKS. This type of reasoning could only work if ELKS depletion would be an all-or-none effect, which is very unlikely to be the case. First, images of ELKS staining in ELKS-depleted cells side-by-side with controls should be provided.

As proposed by the reviewer, we have now added IFM panels of control- and ELKS-depleted cells (**Revised Figure 4e, page 9**). In order to strengthen the specificity of our "home-made" anti-ELKS antibody, we have used in parallel the related pre-immune rabbit serum as another control. We also quantified that the ELKS intensity

of fluorescence is dropped upon its depletion (**Revised Figure 4f**), reflecting the observation made previously by WB. Interestingly in ELKS-depleted cells, we observed residual staining at the cortical sites of the plasma membrane (bottom imaging plane where cells are in close contact with the coverslips, **Revised Figure S4b**), whereas fluorescent staining was dramatically dropped in imaging planes covering the Golgi area (**Revised Figures 4e-f**). This indicates that ELKS proteins might reside in two 'pools' of different half-life: one stable most likely anchored at the plasma membrane and a second much more dynamic (and affected by our siRNA depletion) associated to intracellular structures like the RAB6 vesicles and melanosomes. Altogether, we think that these observations explained the 50% decrease by immunoblotting that can still represent an efficient extinction of the dynamic fraction of ELKS recruited to RAB6 vesicles and thus functioning in the Golgi-melanosome pathway; rather explaining the observed phenotype.

Second, percentage of Rab6 vesicles colocalizing with melanosomes and vice versa, in control and ELKS-depleted cells should be quantified.

As suggested by the reviewer, we have now included this data (**Revised Figure 5d**) that showed a decrease in the colocalization of RAB6 with VAMP7-positive melanosomes in ELKS-depleted cells relative to controls (**page 10**). Note that cells expressing GFP-ELKS- Δ Ct failed to associate RAB6 to donut-like melanosomal structures (**Revised Figure 4j**).

The total number of Rab6 vesicles should also be quantified, since it is expected to increase if ELKS indeed induces a fusion defect.

We quantified the number of RAB6 vesicles in Control- and ELKS-depleted cells and we did not observe significant change in the number of RAB6 vesicles (**Revised Figure 5b, page 10**). This data fit with previous publication showing that ELKS did not impact the formation of RAB6 vesicles (*Grigoriev et al, Dev Cell, 2007*). However our expectation differs from the one of the reviewer, given that RAB6 vesicles undergo kiss&run fusion with melanosomes (see comment above), we are not consuming them totally within melanosomes. Therefore, we are not expecting a dramatic increase in their number upon ELKS-depletion, expectation that fits with the new data.

Further, the reduction of Rab6 vesicle pausing in ELKS depleted cells might be due to a defect in fusion with the plasma membrane, and not with melanosomes, and thus does not provide additional support for the proposed trafficking route.

The reviewer is right. Hence we have now categorized the pause, one close to melanosomes or elsewhere (and in fact, most of those vesicles are very close to plasma membrane). Our additional data (**Revised Table 1, page 26**) now show a reduction of RAB6 vesicles pausing specifically at melanosomes in ELKS-depleted cells.

5. Model in Fig 5 - can the authors exclude that before fusion, ELKS is present

on melanosomes and not on secretory carriers?

The reviewer is correct and indeed we can't exclude this possibility. In parallel, we got unpublished evidences that a minor fraction of ELKS might be already associated to subdomains of melanosomes. This specificity might be achieved by interaction with melanosomal protein. This question is of importance and will keep our attention as a follow-up of this submitted work. Indeed we were already referring to this possibility by mentioning in the discussion (**page 15**) that ELKS might be stabilized at melanosome membranes to constrain RAB6 vesicles to fuse with melanosomes.

Minor comment: although not essential, the authors might consider making a CRISPR/Cas9 knockout of ELKS in MNT-1 cells, since it is likely to be easier and more efficient than protein depletion.

We thank the reviewer for this suggestion. Given that our first siRNA approach was successful in MNT-1 and Human primary cells, and correlated with our in vivo observations in RAB6 KO mice, we proceed with that method. However in the future, we will consider generating KO cells for ELKS alone or in combination with RAB6 using CRISPR/Cas9 technology.

We would like also to thank the reviewer for his/ her insightful comments. We hope that our answers, modifications and clarification will satisfy him/ her.

Reviewer #3 (Remarks to the Author):

Melanosomes are lysosome-related organelles specialized in the synthesis and storage of melanin pigment. How these organelles are put together is a subject of significant current interest. In this manuscript, the authors describe a novel route for transport of a subset of melanosomal proteins directly from the TGN to melanosomes, in addition to the better known route involving transport from endosomes to melanosomes. This direct route depends on the small GTPase Rab6 and its effector ELKS, previously implicated in secretory transport from the TGN to the plasma membrane. This study thus illustrates how a secretory route is adapted to deliver proteins to lysosome-related organelles in a specialized cell type.

The data are generally of high quality. However, the authors need to address (editorially or experimentally) the following issues:

We thank the reviewer for finding our data as high quality.

Fig. 1: The IFM of RAB6 in comparison to melanosomal markers is not very convincing. Can the authors quantify colocalization or "apposition" over the entire cell?

As suggested by the reviewer, we have quantified the apposition of endogenous RAB6 with respect to pigmented melanosome using ICY software plugin – Spot Detector — which allows us to measure the fluorescent intensity of endogenous RAB6 in apposition to pigmented melanosomes. Such analysis was performed on MNT-1 cell line, and primary human and mouse melanocytes in WT or depleted RAB6 conditions. The result now shows that each melanosome is associated in average with 1.5 RAB6-fluorescent structures (**Revised Figures 1c, S2h, pages 5-6**) independently of the cell used.

Further we also quantified the colocalization (Pearson Coefficient) of GFP-RAB6 with mCh-VAMP7 using JacoP ImageJ plugin (**Revised Figure 1d, page 5**). As a control for the specificity of our approach, we performed the same quantification on same images, for which one of the fluorescent channels was rotated (90°) as already described in other studies (*Ryder PV et al, MBoC, 2013*), in that situation, the Pearson Coefficient was dramatically dropped. We add also the similar quantification in ELKS-depleted MNT-1 cells showing the specific loss of co-localization between GFP-RAB6 and mCh-VAMP7 melanosomes relative to control (**Revised Figure 5d, page 10**).

The insets in b-d show colocalization in a swollen vesicle that does not seem representative of a typical Rab6 vesicle. Would it be possible to show colocalization of a set of vesicles instead?

We regret if we were not explicit enough. The insets in the submitted panels Fig1b-d showed tagged-RAB6 (either GFP or mCherry) associated to melanosomes positive for different specific “markers” (like TYRP1, VAMP7 or RAB27A). Thus, RAB6 associated to melanosomes and not to swollen vesicles. In addition to the linescan panels that were already associated to this figure, we have now included a quantification of the colocalization of RAB6 with VAMP7 demonstrating the specific

association of RAB6 to melanosomes (see above and **Revised Figures 1d-e, S1c-d, page 5**).

Fig. 1g: Although I understand that it is difficult to detect endogenous Rab6 by IEM, have the authors tried to detect it using purified melanosomes as in Fig. 1g? Perhaps the use of purified melanosomes affords greater sensitivity and lower background.

We thank the referee for this suggestion and this experiment is now included in **Revised Figures 1h (page 6)**. Endogenous RAB6 staining localized to the limiting membranes of purified melanosomes from MNT-1 cells as well as primary human melanocytes. Based on our biochemical characterization of these fractions (**Revised Figure 1j**), those melanosomes-enriched fractions were devoid of Golgi membranes (GM130) or cytosolic contaminants (GAPDH) reinforcing the specificity of our approach and observations. This panel is put together with anti-GFP labeling performed on purified melanosomes from GFP-RAB6 transfected MNT-1 cells showing that endogenous and overexpressed RAB6 localize similarly.

Fig. 2e: Did the authors see an increase in the number or proportion of immature melanosomes in RPE as they did in MNT-1 cells?

The submitted in vivo experiments were done on adult mice (>4 months) for which the number and proportion of immature melanosomes in RPE cannot be addressed. Previous studies from the group of Clare Futter (*Lopes VS et al, MBoC, 2007*) showed that RPE in adult mice harbored no or very few immature melanosomes in WT mice, but also in mice for which pigmentation were impaired (Rab38^{-/-} mice). Presumably, this would be due to the instability of immature melanosomes (devoid of pigments) over time. Thus we add now a note in the revised manuscript referring to this particular point (**page 16**).

However in order to address this question, we used an ex vivo approach for which RAB6 expression in primary mouse melanocytes can be extinguished upon treatment with 4-hydroxy-tamoxifen (**Revised Figures 2e-f and S2g-k**). The RAB6 KO melanocytes harbored fewer pigmented melanosomes but also increased number of immature melanosomes (**page 6**), corroborating our in vitro data in MNT-1 cells or primary human melanocytes, but also showing in a closest physiological model that RAB6 controls the maturation of melanosomes ex/ in vivo.

We hope that our strategy and new results will satisfy the reviewer.

Fig. 3b: Between 7.1 and 7.9 s, did the Rab6 vesicle approached the VAMP7 vesicle and fused with it, or it just lost its Rab6?

The purpose of the figure was to show RAB6 vesicle approaching and docking to VAMP7 melanosome. This event described one RAB6 vesicle approaching a melanosome that is already associated to one other RAB6 vesicle. The two RAB6 vesicles merged at 7.9s, stayed for a while and then moved away. That represents most likely a kiss&run event. In the revised version of the manuscript, we emphasize much more on the process of kiss&run that underlie most of the dynamic contacts we observed between RAB6 vesicles and melanosomes. For instance, we monitored the

partial transfer of NPY fluorescence from a vesicle to a melanosome, showing a partial mixing of content (**Revised Figures 3h-i** and **Movie S6, page 8**). We also illustrated that RAB6 vesicles that emerge from the Golgi directly contacted a melanosome, stayed for a while, and then possibly reached another melanosome (**Movie S3** and **Revised Figure S3a**). Together, this indicates that RAB6 vesicles can likely undergo successive kiss&run events with distinct melanosomes. We have added additional explanations on these observations (**pages 7-8**).

Fig. 4a: Did siRNA to RAB6 also decrease ELKS? Is the difference reproducible and significant?

We did not observe an alteration of the ELKS expression in RAB6-depleted cells (and vice-versa), as measured in **Revised Figure 4a** (normalized to tubulin expression).

Fig. 4b: The authors conclude that the double RAB6-ELKS depletion is not significantly different from the single depletions, but it's not clear from the figure or the legends what statistical analysis was used to reach this conclusion.

We agree and we regret because the supplementary material and methods files was apparently not available online during our first submission. We have now included additional biochemical experiments showing that melanin decrease was not statistically different in double KD cells relative to single ones (**Revised Figure 4b, page 9**). Additionally, we quantified the percentage of immature and mature stages in single KD (siRAB6 or siELKS) and double (siRAB6+siELKS) KD MNT-1 cells and showed accordingly similar increase of stages-II associated with a decrease of stages-III and -IV (**Revised Figures 4c-d, page 9**).

Fig. 4g: Show siRNA control for this experiment.

As suggested, we have added the siRNA control experiment next to each other in the **Revised Figure 5a**.

Fig. 5a: Why do siRAB6 cells have increased TYRP2 and MART-1 levels?

We showed that RAB6 depletion led to the accumulation of MART-1 at the Golgi area (MNT-1 and primary human melanocytes; **Revised Figures 6f-h** and **S5e**). At least one explanation is that the sorting of MART-1 away from the Golgi apparatus is defective leading to its accumulation. Unfortunately, parallel experiments following endogenous TYRP2 cannot be performed due to the lack of available antibody suitable for immunofluorescence. Although not tested, we cannot exclude that some transcriptional up-regulation specifically of these two cargoes might occur upon RAB6 depletion. However and even if such gene regulation occurs, only fewer MART-1/ TYRP2 proteins reach melanosomes, as revealed biochemically by their specific decrease in the melanosome-enriched fractions of RAB6-depleted cells (**Revised Figure 6c**) identifying a sorting/ trafficking defect from Golgi to melanosomes as an explanation (**page 11**).

Fig. S5b: It's probably not necessary to show these data since they concern a chimeric protein that is not used in further experiments.

We understand this point, however we would like to keep that figure as it is for the following reason. We have generated the full length GFP-TYRP2 construct, however this version did not localize properly (massive retention in the endoplasmic reticulum). Therefore we have generated a truncated version corresponding to the transmembrane and cytosolic domains (containing most likely the sorting motifs) of TYRP2 in fusion with GFP. Since we have used this chimera to identify its localization in MART-1 vesicles by EM (**Figure 6e**), we thought it was important to show that this chimera was also properly targeted to melanosomes (**Revised Figure S5d**) like the full length should do. Thus, this panel corresponds to a control for the specificity of this newly generated construct (that may be helpful for colleagues, if interested).

We would like to thank the reviewer for having critically reviewed our manuscript and we hope that our answers and additional experiments will satisfy his/ her comments.

Reviewer #4 (Remarks to the Author):

The paper by Patwardhan et al entitled "Routing of the Rab6 secretory pathway towards lysosome related organelles" investigates the possible function of Rab6 and ELKS in the biogenesis of melanosomes. The authors use a variety of approaches, present numerous results and make some interesting observations. Overall the data suggest a function for Rab6 and ELKS in the biogenesis of melanosomes. However, the data presented does not satisfactorily support the claim of a direct route from the TGN to melanosomes that is dependent on Rab6/ELKS and used by Tyrp2 and MART-1 but not by Tyr and Tyrp1.

We thank the reviewer for finding our study interesting and for appreciating our experimental approach and we hope that our answers will satisfactorily address his/her concerns.

Major points:

(1) The localization of Rab6 to melanosomes is not adequately supported by the data.

First, the localization of overexpressed GFP-Rab6 differs from that of endogenous Rab6 detected by immunofluorescence microscopy. Endogenous Rab6 does not show a "doughnut" type staining around the melanosome as observed for GFP-Rab6 suggesting GFP-Rab6 overexpression produces an artifact.

The reviewer refers here to the differences observed by IFM between the localization of endogenous RAB6 (punctate structures associated to melanosomes; revealed by melanin or melanosomal markers), and the typical 'doughnut'-like melanosome staining of exogenous (GFP/mCherry-RAB6) that overlapped almost completely with melanosomal markers. While these observations revealed a difference in the abundance of RAB6 (endogenous vs. exogenous) to melanosomes, they demonstrated that both endogenous and exogenous RAB6 associated with melanosomes.

Now, the endogenous localization of RAB6 was reinforced by showing similar observations in primary human and mouse cells using different experimental setups (IFM, biochemistry and EM on cell sections or isolated melanosomes). In average, we quantified 1.5 endogenous RAB6-positive fluorescent structures associated per melanosomes in all cells tested. By IEM on isolated melanosomes from primary human melanocytes or MNT-1 cell line, endogenous RAB6 staining associated with the limiting membranes of melanosomes as for GFP-RAB6. Similarly, biochemical analysis of the melanosome-enriched fractions showed the presence of endogenous RAB6 and GFP-RAB6 (**Revised Figures 1, S2, pages 5-6**).

In addition, by expressing a fluorescent intra-body recognizing endogenous activated RAB6 (RAB6:GTP; *Nizak et al, Science, 2003*), we now show by live cell imaging that RAB6:GTP localized to discrete structures associated with melanosomes (**Revised Figure S1b and Movie S1**) as in fixed cells. Note also, that the intra-body can also co-localize almost perfectly with a doughnut-like VAMP7-positive melanosome (**Revised Figure S1b**), as overexpressed RAB6. Exogenously,

swapping the GFP tag for mCherry did not change the melanosomal localization of exogenous RAB6 (**Revised Figure 1, S1**). Also the association of GFP-RAB6 to melanosomes depends on its active state, because GTP-locked GFP-RAB6 version localized to melanosomes but not the GDP-locked form (**Revised Figures S1e-f, page 5**).

Finally and importantly, melanocytic cell lines expressing GFP-RAB6 but depleted for ELKS decreased the co-localization of RAB6 with VAMP7 at melanosomes (**Revised Figures 5a, 5c-d**). Such observation was confirmed when mCh-RAB6 was co-expressed with a RAB6 binding-deleted version of ELKS (GFP-ELKS- Δ Ct, **Revised Figure 4i-j** and **Movie S7, page 10**). This shows that the association of RAB6 to melanosomes requires its interaction with ELKS and thus, altogether, does not represent fortuitous localization but rather a specific one.

The difference in the visible localization (dotty structures (endogenous) vs. donut structures (exogenous)) reflects most likely the abundance of RAB6 associated to melanosomes and not an unspecific association. We can hypothesize that excess of RAB6 might impact the equilibrium between cytosolic and membrane-associated RAB6, leading to more RAB6 associated to melanosomes, perhaps by saturating a specific RAB6 GAP at melanosomes. In such case, RAB6 might be stably localized to melanosomes but not unspecifically targeted.

All together, we hope that our explanations and additional experiments are now reinforcing our data and addressed the point raised by the reviewer.

Second, Figure 1h (top part) is problematic. The experiment uses cells transfected with GFP-Rab6, which are then lysed, subjected to melanosome enrichment, IP with an anti-GFP antibody, and WB analysis with various antibodies. If GFP-Rab6 localizes to melanosomes, the "IP" lane should show Tyrp1 but it does not. This goes directly against the idea that GFP-Rab6 is present in melanosomes and all the conclusions made from fluorescence and electron microscopy data involving GFP-Rab6.

We are sorry because our description was maybe misleading. From a technical point of view, we mechanically broke the cells (and not lysed them), purified pigmented melanosomes based on their density using Sucrose flotation, collected the melanosome-enriched fraction and then lysed using immunoprecipitation buffer (containing NP-40 detergent). This fraction, referred as Mel input, was incubated with GFP-trap Agarose beads in order to perform our GFP immunoprecipitation. Also we disagree that the 'IP' lane should show TYRP1 (see below).

Regarding our first biochemical characterization (first submission), using such a protocol, we first verified the quality of the "Mel Input" fraction by testing it for any contaminant. This fraction was devoid of Golgi membrane associated proteins (GM130) precluding RAB6 detection from the Golgi source. Then we detected specific enrichment of melanosomal components like TYRP1 proteins showing that the Mel Input fraction was indeed enriched in pigmented melanosomes. Note that we have also tested this fraction (before lysis) by EM and used it for the endogenous RAB6 immunolabeling shown in **Revised Figure 1h (page 6)**. Then we performed the GFP immunoprecipitation on the lysed Mel Input fraction and had detected the GFP-RAB6 corresponding band. However and as opposed to the reviewer, we are

not expecting that the GFP-RAB6 associated to melanosomes interact with all melanosomes-associated components, like possibly TYRP1. This is not because they co-distribute that they must interact. Our observation showing that TYRP1 was not particularly enriched in our immunoprecipitated lane suggested that RAB6 does not interact with TYRP1.

Now in the revised manuscript and for as sake of clarity, we have now replaced this panel by showing side by side the association of endogenous and exogenous RAB6 in the melanosome-enriched fraction of primary human melanocytes and MNT-1 expressing or not GFP-RAB6 (**Revised Figure 1j, page 6**), but without performing immunoprecipitations. This panel demonstrates the association of endogenous RAB6 with melanosomes fractions (Mel and additionally revealed by IEM) also positive for melanosomal markers (TYRP1, TYRP2/DCT and MART-1). Those fractions were devoid of resident Golgi or cytosolic proteins (GM130, GAPDH) found in the post-nuclear supernatant (PNS), confirming that endogenous RAB6 did not correspond to co-fractionation of Golgi membranes cytosolic contamination with melanosomes but rather could be derived from either RAB6 vesicles or RAB6 already bound to melanosomes as proposed by the model.

Likewise, if GFP-Rab6 and endogenous Rab6 localize to the same compartment, the "IP" lane should show endogenous Rab6 but this is not the case. It appears that GFP-Rab6 does not appropriately represent or localize as endogenous Rab6. This is a significant problem.

Endogeneous RAB6 would be present in the IP lane if GFP-RAB6 interacts with endogenous RAB6 (in other words if they homo-dimerize). To our knowledge, RAB6 was never shown to dimerize. Therefore, GFP-RAB6 localizes appropriately as endogenous RAB6. Also we are now showing that endogenous or exogenous RAB6 associate to the limiting membranes of melanosomes (isolated from primary human melanocytes and MNT-1 cells, **Revised Figure 1h**). Isolated melanosomes that were also examined biochemically were positive for endogenous and exogenous RAB6 (**Revised Figure 1j**). This altogether supports that endo/ exogenous RAB6 localize similarly to melanosomes (**pages 5-6**).

Third, these cells are overexpressing GFP-Rab6, therefore it is not clear if the ~23KDa band observed is truly endogenous Rab6 or the product of GFP-Rab6 cleaved during the preparation of cell lysates.

We have taken into account this point. The revised version of this figure now shows the presence of endogenous RAB6 in the melanosome fraction and thus rule out this possibility. We have also compared endogenous RAB6 and GFP-RAB6 in melanosomal fraction and shows indeed the enrichment of RAB6 in melanosomal fractions, independently of the GFP-tag (**Revised Figure 1j, page 6**). Further we also demonstrate that endogenous RAB6 associated with purified melanosomes.

Additionally, the presence of the Rab6 band in the "Mel input" lane is a weak evidence that the protein is indeed in melanosomes because the enrichment process is rather crude and this preparation likely includes other membrane bound compartments. These concerns could be partially alleviated if immuno

EM for endogenous Rab6 rather than overexpressed GFP-Rab6 were to show the protein localizes to pigmented melanosomes. This immuno EM analysis should include quantification and negative controls, such as melanocytes from the KO mouse or siRNA treated MNT-1 cells. The biochemical experiment should also be performed with control, non-transfected cells, and both immunoprecipitation and WB analysis should use antibodies to endogenous proteins (Rab6, Tyrp1, etc). These experiments could support the premise that endogenous Rab6 localizes to pigmented melanosomes. Still the question would remain regarding the validity of using overexpressed GFP-Rab6.

We agree on this aspect and have provided additional data. As proposed by the reviewer, we demonstrate now by western blots the association of endogenous RAB6 to enriched melanosome fractions (positive for TYRP1, TYRP2 or MART-1 and free of cytosolic/ Golgi membranes contaminations, **Revised Figure 1j**) of primary human melanocytes as well as MNT-1 cells expressing or not GFP-RAB6. Most importantly, this result is further confirmed by anti-RAB6 immuno-Gold labeling of the same biochemical fractions that shows RAB6 association to the limiting membranes of those melanosomes (**Revised Figure 1h-i, page 6**). Those experiments showed that endogenous and exogenous RAB6 associate to melanosomes.

By immuno-EM (**Revised Figure 1h**), we can first reveal that the melanosome-enriched fractions are not crude, as also revealed by immunoblotting (**Revised Figure 1j**). Second, we counted the number of anti-GFP gold particles associated to melanosomes (expressed as % of gold labeling on stages-III or -IV pigmented stages, **Revised Figure 1i**). Third, we have evidenced the specificity of that GFP-RAB6 staining because labeling the same preparation using only protein-A-gold (10nm, data not shown) or anti-GFP antibody (in GFP-expressing melanocytes, **Revised Figure 1h**) did not show gold particles associated to melanosomes reinforcing the specificity of our RAB6 labeling approach.

Performing the same experiments and analysis on purified melanosomes from RAB6 KO melanocytes or KD melanocytes might give wrong estimates or interpretation due to the lack of RAB6 epitopes (KO) or remaining RAB6 proteins from non-depleted cells (KD is non homogenous. This is why we have favored the addition of control of the specificity of binding of the RAB6 or GFP antibodies.

Together with our new data and comments above regarding the specificity of the overexpression approach, our data show now convincingly that RAB6 associate with melanosomes (**pages 5-6**).

(2) The reduced pigmentation and biogenesis of pigmented melanosomes in Rab6 (and ELKS) deficient cells is convincing although no picture of the Rab6 KO mice was shown in order to observe if there is a fur or eye pigmentation defect (this should be provided).

We are currently characterizing in depth the phenotypes associated to the RAB6 KO mice. Apart from the defect in the biogenesis of melanosomes in RPE cells that we illustrate in the manuscript, these mice suffered from other defects that are under investigation. These results will be the part of a future complete study focusing on the physiological role of RAB6 during pigmentation.

Our present study deciphers the 'molecular' function of RAB6 in melanocyte with an

in vivo emphasis using pigmentation in RPE. In vivo, the pigmentation and biogenesis of melanosomes in the RPE are known relevant models to illustrate the in vitro pathways required for the biogenesis of melanosomes in skin melanocytes (Rochin et al, PNAS, 2013; van Niel et al, Cell Rep, 2015; Lopes et al, MBoC, 2007). The in vivo data presented in this manuscript highlighted a pigmentation defect in the eyes of those mice associated to a statistical decreased of the number of pigmented melanosomes in the RPE (**Revised Figure 2**) that fully support our in vitro characterization. Moreover we add now data on primary mouse melanocytes isolated from RAB6 KO inducible mice (Bardin et al, 2015, Biol Cell). RAB6 extinction recapitulated ex vivo our in vitro data (**Revised Figures 2e-f and S2g-k, page 6**).

However, this result does not discriminate between a defect in transport specifically to melanosomes from an indirect effect of disrupting general secretory transport.

The reviewer is correct. Given that RAB6 might control the Golgi homeostasis in melanocytes, we have now provided additional data regarding Golgi functions. A disruption of Golgi functions might lead to general defect impacting globally transmembrane protein glycosylation and/ or trafficking.

By immunoblotting of the lysates of RAB6-depleted cell, we did not observe change in the migration patterns of TYRP1, TYR or PMEL (three heavily glycosylated transmembrane melanosomal proteins), indicating no general defect on their glycosylation and thus no global malfunctions of the Golgi (**Revised Figures S5a-c, page 11**). In addition, we showed that PMEL (which undergoes secretion to cell surface and then re-internalization and transport towards stage-I melanosomes (Bissig et al, Int J Mol Sci, 2016)) is properly processed (between Golgi and endosomes) as illustrated by the similar amount of PMEL-N/C (**Revised Figure S5b**) and of the HMB45-positive PMEL fibrils (see below). In vivo, remaining melanosomes in RAB6 KO mice have similar shape as in control (**Revised Figure 2, S2**) indicating that the formation of PMEL fibrils is not affected (Rochin et al, PNAS, 2013; van Niel et al, Cell Rep, 2015). Together our data indicates that the loss of RAB6 expression does not impact Golgi functions (sorting, trafficking or homeostasis).

-Second, no general Golgi sorting defect (apart for MART-1 sorting and most likely TYRP2) was observed since we did not detect alteration in TYRP1 localization, by IEM on RAB6-depleted MNT-1 cells (**Revised Figure S5f**).

Altogether our new data support a specific role of RAB6 in the delivery of particular components (MART-1 and TYRP2) directly from Golgi to melanosomes.

Also, Rab6 siRNA cells show many stage I but no stage II melanosomes can be observed in the figure. This is also the case for ELKS deficient cells. Is this true? Stage I and II melanosomes should be further discriminated in the quantification. This is important because it appears that the problem is in the transition between stage I and II melanosomes rather than between stage II and III as proposed by the authors.

The reviewer is right and we are sorry because the chosen pictures could be misleading. In both conditions (RAB6- and/ or ELKS-depleted cells), we have now performed the suggested quantification to show the data for all four stages. While the number of stage-I increased slightly (but not significantly), we quantified much more stage-II and a concomitant decrease in the number of stages-III and -IV. These quantifications were performed in RAB6 KO inducible mouse melanocytes (relative to non-induced ones), as well as Control-, RAB6-, ELKS- and RAB6+ELKS-siRNA treated MNT-1 cells.

This data supports a stage-II to -III maturation defect in absence of either RAB6 or ELKS. Thus we have modified accordingly the **Revised Figures 2, 4 and S2 (pages 6 and 9)**.

Pmel17 should be analyzed to better establish if there is a problem in generating stage II melanosomes. The HMB45 antibody recognizes the Pmel17 fragments present in the fibrils of stage II melanomes and should be compared in WB analysis and immuno EM of control, Rab6 and ELKS deficient cells. This experiment may help discriminate between a specific transport defect to stage II melanosomes (as proposed by the authors) from a non-specific defect in general secretion. Pmel17 does traffic, at least in part, through the plasma membrane "en route" to melanosomes and could very well be affected by the Rab6/ELKS deficiency. The authors should explore the possibility that Pmel17 transport and processing is defective.

As proposed by the reviewer, and already detailed above, PMEL expression, processing and fibrillation were explored in MNT-1 cells and no defect was observed as compared to controls (**Revised Figure S5b, page 11**). In addition to the provided quantification of all stages, our data supports a role for RAB6 and ELKS on a specific transport route between Golgi and pigmented maturing melanosomes.

(3) Do GFP-Rab6 labeled vesicles observed in Fig 3b actually fuse with the Vamp7 compartments? If so, what proportion of them does fuse?

The data in Figure 3c shows a GFP-Rab6 labeled vesicle apparently fusing with a compartment, which is not labeled but presumably would be a melanosome. This organelle should be identified as a melanosome by simultaneously imaging the pigment or a melanosome marker such as the one used in Fig 3b, Vamp7. It is hard to see whether there is a fusion event or the compartments are getting in/out of the focal plane. Also, the structure indicated as a vesicle in Fig 3c seems too big for a vesicle. A reference size bar should be included for comparison.

The reviewer is absolutely right that we don't identify the ring-like structure as a melanosome. In fact, we have assumed it, due to its size and shape by optical microscopy. This is also why we were cautious in the submitted text by saying (page 7, line 161-164) "*RAB6 fluorescence disappearing from the docked vesicles (arrows) increased concomitantly at ring-like structures (arrowheads) (Figure 3c and Movie S5), most likely indicative of fusion events occurring between RAB6 vesicles and melanosomes.*"

However we have realized that we were not enough clear about the frequency and different types of fusion events that were observed between RAB6 vesicles and melanosomes. Our careful examination of RAB6-positive dynamics relative to melanosomes indicate that they fuse via kiss&run mechanisms rather than a complete fusion and consumption of vesicles into melanosomes. We now show that RAB6 vesicles emerging from the Golgi directly contact a melanosome, stay for a while, and then possibly meet another melanosome (**Movie S3, page 7**). This explain why, and even if the movies were acquired with 10 frames / sec, no complete obvious fusion of RAB6 vesicles were recorded. Also, given that GFP-RAB6 associated with melanosomes in addition to the vesicle, it challenges such observation. Together, this indicates that RAB6 vesicles can likely undergo successive contacts and kiss&run events with distinct melanosomes. In the revised manuscript, we have removed this 'fusion panel' (submitted Figure 3c and movie S5), first to avoid confusion and second to reinforce the plausible suggestion that RAB6 vesicles undergo kiss&run events with melanosomes.

The kiss&run events are now illustrated by the measure of the NPY-Venus fluorescence intensity (normalized to the one of mCherry-VAMP7) that showed a raise of NPY intensity within a melanosome at the time where the NPY-vesicle establish tight contact with the VAMP7-positive melanosome (**Revised Figures 3h-i and Movie S6**). This data illustrates the partial transfer of NPY from the vesicle to the melanosome and thus a kiss&run event. In addition to the IEM showing NPY in the lumen of pigmented melanosomes (**Revised Figures 3e**) and the **revised Movie S3**, our data indicate that one vesicle can undergo multiple and successive kiss&run events with different melanosomes (**page 8**).

Knowing that most of the NPY vesicles co-localized with RAB6 (new quantification provided in **Revised Figure S3b-c**), and that RAB6 dynamic contacts and associations with melanosomes are decreased upon ELKS depletion or the expression of its RAB6 binding mutant (**Revised Figures 5, 4i and Movie S7**), we believe that we provide altogether a body of evidence indicating that RAB6 vesicles fuse with melanosomes through kiss&run.

Therefore, we now amended this point by slightly modifying the text in order to reveal that kiss&run behaviour is the most common observations (**pages 7-10**).

(4) Since many NPY-Venus vesicles are not labeled by Rab6 (Fig S3c), it is unclear if the NPY-Venus delivered to melanosomes (Fig. 3f-h) was actually contained in Rab6 labeled vesicles or if Rab6 was needed for such transport.

RAB6 localize abundantly to the Golgi apparatus and to vesicles that follow the anterograde and retrograde routes (Golgi-plasma membrane and endosome-Golgi, Golgi-endoplasmic reticulum). Thus, we do not expect a total overlap between RAB6 and NPY, as shown in previous studies (60% of the RAB6- and NPY-positive

vesicles; *Grigoriev et al, Dev Cel, 2007*), We are now providing quantification for RAB6 co-localization with NPY in MNT-1 cells (Pearson correlation coefficient 0.6 ± 0.02 , **Revised Figure S3b-c, page 8**). As expected from previous studies in non-pigment cells showing similar RAB6/ NPY co-localization (*Grigoriev I, 2007, Dev Cell*), this data shows a significant fraction of the RAB6-Golgi derived vesicles that indeed associates with the exogenously expressed secretory cargo NPY in melanocytes.

We would like to stress here what was the rationale of using NPY in our model system. In order to monitor the dynamic exchange of content from RAB6 vesicles to the melanosomes, we looked for a soluble reporter molecule packaged specifically into RAB6 vesicles. The pathway we described corresponds to a post-Golgi route that ends in melanosome. Until now, the main RAB6-dependent post-Golgi route described so far is the Golgi-Plasma Membrane secretory pathway. Given that NPY is a canonical cargo of this RAB6 secretory pathway, we used NPY-Venus as a soluble and fluorescent reporter molecules allowing us to track possible fusion events between post-Golgi RAB6 vesicles and melanosomes. Thanks to this “artificial system”, we were able to monitor partial exchange of fluorescence from NPY-positive vesicles to melanosomes (**Movie S6** and **Revised Figures 3h-i**), as well as observing NPY-Venus staining within the lumen of pigmented melanosomes (**Revised Figure 3e**). This result led us to propose that RAB6 vesicles can fuse with melanosomes and that the Golgi-Melanosome pathway might share properties of the secretory pathway or might correspond to its adaptation. Later our hypothesis was reinforced by the implication of ELKS in that pathway.

(5) Biochemical studies with GFP-ELKS (Fig S4C) have a similar problem as described above for GFP-Rab6. The "IP" lane shows enrichment of GFP-ELKS relative to the "Mel Input" lane, however little Tyrp1 and Tyrp2 is present in the "IP" lane compared to the "Mel Input" and "Unbound" lanes. This result indicates the bulk of the overexpressed GFP-ELKS is not present in pigmented melanosomes.

We are sorry again if our explanation was not clear in the submitted manuscript, but we do not agree with the interpretation of the reviewer. The answer regarding the biochemical association of RAB6 to melanosomes and detailed above (major point#1 of the same reviewer) applies similarly here.

We have modified accordingly our answer (see above) to specifically address that concern.

Regarding our first biochemical characterization (first submission), from a technical point of view, we mechanically broke the cells, purified pigmented melanosomes based on their density using Sucrose flotation, collected the melanosome-enriched fraction and then lysed (immunoprecipitation buffer containing NP-40 detergent). This fraction, referred as Mel input, was incubated with GFP-trap Agarose beads in order to perform the GFP immunoprecipitation. Using such a protocol, we first verified the quality of the “Mel Input” fraction by testing it for contamination. This fraction was devoid of Golgi membrane associated proteins (GM130) precluding detection from other sources. Then, we detected specific enrichment of melanosomal components like TYRP1 or TYRP2 proteins showing that the Mel Input fraction was indeed enriched in pigmented melanosomes. Then we had performed the GFP immunoprecipitation on the Mel Input fraction and had detected the GFP-ELKS

corresponding band. However, we are not expecting that the GFP-ELKS associated to melanosomes interact with all melanosomes-associated components. This is not because they co-distribute that they must interact. Our observation showed that a fraction of the melanosome-associated TYRP1 and TYRP2 were detected in the IP lane, possibly suggesting that ELKS could interact with these two components.

Now in the revised version and to clarify and strengthen this particular data, we have replaced the original figure by a similar experiment probed only for endogenous components. Accordingly to our first observation, endogenous ELKS associated to the melanosome-enriched fraction of MNT1 cells (**Revised Figure 4h, page 9**). We hope that our collective explanation (here and to first point above) makes clear that a fraction of endogenous RAB6 and ELKS associate with pigmented melanosomes.

Consequently, it appears that organelles visualized in microscopy experiments with GFP-ELKS (Fig 4e-g and S4a,b,d) for the most part are not melanosomes. This issue is exacerbated by the lack of quantification of these microscopy experiments. As discussed above for Rab6, this is a significant problem because it goes against the microscopy data obtained with the same GFP-ELKS construct.

According to our answers to these specific points, we do not think that our biochemical data goes against the microscopy data. On the contrary, they fully support the association of a fraction of endogenous RAB6 and ELKS to melanosomes. In addition, we have now improved our immunofluorescence data on endogenous ELKS and showed its close association with CD63 (a tetraspanin enriched at pigmented melanosomes, **Revised Figure 4e**, *van Niel et al, Dev Cell, 2011*).

We have now added IFM panels of control- and ELKS-depleted cells (**Revised Figure 4e, page 9**). In order to strengthen the specificity of our “home-made” anti-ELKS antibody, we have used in parallel the related pre-immune rabbit serum as another control. We also quantified that the intensity of fluorescence is dropped upon ELKS depletion, reflecting the observation made previously by WB.

In melanocytes, ELKS localize to plasma membrane structures (as already described in several studies), but also to RAB6-positive vesicles and melanosomes. Thus we can only expect a partial overlap of ELKS staining with melanosomes (as shown endogenously by IFM and confirmed by immunoblotting on fractionation). By live cell imaging, we show now that GFP-ELKS associated to dynamic RAB6 vesicles emerging from the Golgi area (**Revised Figure 4i, Movie S7**). However GFP-ELKS was barely detectable at the Golgi per se. Given that ELKS is described as a RAB6 effector (*Monier, 2002, Traffic*), this indicates that ELKS might be recruited by RAB6 on post-Golgi-derived vesicles. Indeed, a deleted Cter version of ELKS (that lacks its RAB6 binding domain) did not co-distribute with RAB6 vesicles and prevent RAB6 localization to melanosomes (**Revised Figures 4i-j, Movie S7, pages 9-10**).

Altogether and in association to the previous explanations, we hope that our additional data convince the reviewer about the specificity of association of RAB6 and ELKS to melanosomes.

(6) It is interesting that Tyrp2 and MART-1 (but not Tyr and Tyrp1) steady state

levels are increased in Rab6 depleted cells. The authors interpret this result as suggesting Tyrp2 and MART-1 use a different pathway to target melanosomes and claim that such pathway is a novel, direct, Rab6-dependent route from the TGN to melanosomes. The data simply does not support such claims. Again this could be an indirect effect of generally altering normal exit from the Golgi apparatus. For instance, the data does not rule out traffic of Tyrp2 and MART through early endosomes. These claims are therefore an overstatement that should be eliminated or corrected, including in the abstract.

The reviewer is correct and this is why we have now added new data to strengthen this particular section. As discussed above, we have excluded a possible Golgi dysfunction upon RAB6 depletion by examining either the overall biochemical migrating patterns of TYR, TYRP1 and PMEL or their localization (TYRP1). Those proteins are three melanosomal components that cross the Golgi and reach the melanosomes through an endosomal intermediates. We did not observe any evidence supporting glycosylation or sorting/ trafficking defects (see above and **Revised Figure S5, page 11**). As well, MART-1 and TYRP2 biochemical migration patterns were similar in RAB6-depleted cells relative to controls, suggesting here also no post-translational modifications.

To rule out that MART-1 or TYRP2 could traffic through the endocytic pathway, we have added the localization of TYRP2 and MART1 together with internalized fluorescent Transferrin. No significant co-distribution of GFP-TYRP2 or endogenous MART-1 with internalized Transferrin was detected (**Revised Figure S5d, page 11**) that, in addition with our IEM showing GFP-TYRP2 and/ or MART-1 together with RAB6 (**Revised Figures 6d-e**), supports a non-endocytic route that is distinct from the one followed by the other known components.

Additionally, quantification of microscopy localization data should be presented (Fig 5c-d).

As suggested by the reviewer we have quantified the number of vesicles positive for either MART1 and RAB6 or MART-1 and GFP-TYRP2, showing that a significant fraction of post-Golgi-derived vesicles were indeed co-positive (**page 11**). 64% of RAB6 vesicles (74 out of 115; 5 Golgi, n=7 cells) co-stained with MART-1, while almost all MART-1 and TYRP2 vesicles were co-stained (105 vesicles; 11 Golgi, n=8 cells). This data indicates that MART-1 and TYRP2 are most-likely co-package in the same RAB6-positive vesicles.

(7) Related to point (6). If Rab6 deficient cells have significantly less stage III/IV melanosomes and the steady state levels of Tyr and Tyrp1 are not affected, where are Tyr and Tyrp1 localized in these cells? It is quite possible that they are retained in the Golgi apparatus due to a general secretion defect. If this were true it would undermine the idea of a separate pathway to melanosomes and would also explain the pigmentation phenotype observed here.

The localization of TYR cannot be addressed by IEM or IFM due to the lack of suitable antibodies for human cells. We have addressed the localization of TYRP1 in Control- and RAB6-depleted cells. TYRP1 did not accumulate in the Golgi apparatus, but still associated to remaining pigmented melanosomes in RAB6-depleted MNT-1

by IEM (**Revised Figure S5f**). These observations are in line with our other data supporting a new and distinct route for MART1 and TYRP2 to melanosomes.

The authors need to compare in a quantitative fashion the localization of Tyr/Tyrp1 in control and Rab6/ELKS deficient cells to establish if their transport is affected or not.

We have now provided biochemically that, upon RAB6 KD, TYRP1 amount in the melanosome-enriched fractions was not affected, while MART-1 and TYRP2 were decreased (**Revised Figure 6c**). Note also that the overall expression of TYRP1 is not impacted as compared to MART-1 and TYRP2 that were increased. We also add IFM data in primary human melanocytes depleted or not for RAB6 that recapitulate the previous observation made in MNT-1 (the accumulation of MART-1 at the Golgi) (**Revised Figure 6f-g, S5e**). Even if we cannot totally exclude that the trafficking of TYRP1 is 100% correct upon RAB6 KD, our data shows that MART-1 and TYRP2 expression, sorting and association to melanosomes are specifically and dramatically affected, together with the fact the melanosomal cargoes trafficking through endosomes are well glycosylated (TYR, TYRP1, PMEL) and processed (PMEL).

Thus, we did not obtain evidence proposing that RAB6 might control the endosome to melanosome pathway.

(8) It is conceptually difficult to understand how two proteins that are ubiquitously expressed (Rab6 and ELKS) mediate transport to a tissue specific organelle (melanosome). How is the specificity achieved?

The reviewer raises an interesting point. How do specialized cell types achieve specific tasks by using a set of ubiquitously expressed proteins? We, and others, already described that ubiquitous trafficking components (AP-1, AP-3, BLOC-1, -2, -3) specifically control the biogenesis of melanosomes. Their functions are indeed revealed at the organism level by showing in vivo, more or less severe hypopigmentation.

This “philosophical” extremely interesting question is the major challenge that highly specialized cells (including melanocytes but not only) have to face. How to ensure their own specificity, while using/ exploiting similar transport routes, same carriers and identical trafficking/ sorting machineries as other (perhaps) less specialized cells? For instance in neurons, how are specific receptors delivered to the axon or dendrites? Many studies point out that specific sorting of cargoes, their interactions with molecular motors or modifications of the cytoskeleton contribute to this specificity of trafficking. We believe, given our previous studies and those by others, that such general assumption might also be applied to the melanocytes.

Regarding our study, we would like to comment on the similarities and differences we observed between the Golgi-melanosomes pathway and the current knowledge on the conventional secretory pathway (Golgi-plasma membrane).

Similarly to the secretory pathway, we did not observe a change in the number of RAB6 vesicles formed upon ELKS depletion (**Revised Figure 5b**), corroborating previous data (*Grigoriev et al, 2007, Dev Cell and 2011, Cur Biol*). As in Grigoriev et al (2007), ELKS-ΔCt-expressing cells did not abolish the transport of RAB6 vesicles

towards the periphery (**Revised Figure 4i**, note the accumulation of RAB6 fluorescence at the cell periphery). Also, depleting ELKS decreased the pause duration at melanosomes (**Table 1**), as shown at plasma membrane (*Grigoriev et al 2007*).

However, some differences were highlighted by our study. (1) Compared with the findings by Grigoriev et al (*2007 and 2011*), ELKS associates with dynamic RAB6 vesicles emerging from the Golgi (**Revised Figure 4i, Movie S7**). (2) In contrast to secretory cargoes (i.e. NPY), RAB6 in melanocytes operates at a step impacting the sorting of melanosomal cargoes (i.e. MART-1) away from the Golgi (**Revised Figures 6f-h and S5e**). (3) While RAB6 depletion delays the secretion of cargoes at plasma membrane (more or less dramatically depending on the cargo that is monitored (*Grigoriev et al 2007 and 2011*)), it strongly impairs the delivery of melanosomal cargoes to melanosomes, as revealed by melanin decrease, maturation defect of melanosomes and decrease amount of MART-1 and TYRP2 to melanosomes (**Revised Figures 2, 4 and 6c**). (4) Similarly ELKS that is not required for the secretion per se (*Grigoriev et al 2011*), tightly controls in melanocytes (as RAB6 did) the transport and fusion of RAB6 vesicles with melanosomes (**Revised Figure 4**).

These results show that the Golgi-melanosomes pathway shares only some features with the “conventional” secretory pathway and thus represents an adaptation/exploitation of it. The underlying mechanisms and machineries leading to this adaptation most likely represent the key to understand its specificity and are currently under investigation. Our results indicate that the specificity is achieved at the Golgi level (via RAB6 for sorting cargoes), at the transport step (via RAB6 and ELKS for contacting melanosomes) and at docking step (via ELKS for stabilizing the vesicle-melanosomes interaction).

By delivering selected melanosomal cargoes to this lysosome-related organelle, this novel pathway is thus a specific mechanism that does segregate between plasma membrane components to be secreted and melanosomal cargoes. Hence this ensures an advantage by promoting a tuning of melanosomes maturation without losing their identity by adding ‘contaminating’ components. And thus there is certainly a real advantage for the melanocytes to tune/ adapt this conventional pathway in a specific route. The NPY experiments are here a ‘trick’ that we have used to follow the fusion of RAB6 vesicles with melanosomes based on the fact that melanocytes do not express endogenous NPY.

We have added a note in the Discussion section and slightly modified the final model that summarizes the differences/ similarities with the secretory pathway to reinforce the idea of its exploitation (**Revised Figure 6h and Page 15**).

Minor points:

Title: this work explores melanosomes rather than multiple lysosome related organelles, therefore “lysosome related organelles” should be replaced with “melanosomes”.

The reviewer is right therefore we have changed the title as follows:

“Routing of the RAB6 secretory pathway towards the lysosome-related organelle of melanocytes”.

Abstract: "Our data together reveal for the first time that the secretory pathway can be directed towards intracellular organelles to ensure their biogenesis and function". Most secretory granules, which are intracellular organelles, derive from the secretory pathway. What is the novelty here?

The novelty resides in the fact that melanosomes originate from the endocytic pathway and not from the Golgi apparatus, such as secretory granules. To our knowledge, no other intracellular organelles have been described to intercept the RAB6/ ELKS-dependent secretory pathway. Secretory granules originate from the Golgi apparatus and thus receive materials from the Golgi apparatus, but, and even if they are secreted like the Weibel Palade Bodies or insulin granules for instance, there are no evidence for such an adaptation of the conventional RAB6/ ELKS-dependent secretory pathway, or even for a role for RAB6 or ELKS, for delivering materials to the maturing secretory granules as depicted in our study.

In multiple occasions the text refers to Rab6 when in reality the experiment was performed with overexpressed GFP-Rab6. Given the concerns mentioned above this has to be clearly stated.

We have corrected the text accordingly.

Figure 4g, the cell seems to have very low levels of GFP-Rab6. Is this a general issue in siELKS cells or just this example?

We have not identified any effect on RAB6 expression upon ELKS depletion. We have performed quantification of number of RAB6 vesicles in Control / ELKS depleted condition and do not show significant alterations in number (**Revised Figure 5b**), nor levels by WB (**Revised Figure 4a**).

Figure 5g, the data is too preliminary to adequately support the model presented. The model should be eliminated.

We think that in line with our explanation and additional experiments, our data does support the proposed model. We believe it is important to keep this model in order to help the reader to understand the role of the RAB6-ELKS pathway in melanocytes and its role in cargo trafficking step as compared to the conventional secretory pathway. Thus, we have slightly modified the final model to illustrate the two pathways originating from the Golgi apparatus.

We would to thank the reviewer for having carefully read and comment our manuscript. We hope that our answers and new data will satisfy him/ her.

Reviewers' Comments:

Reviewer #1:

Remarks to the Author:

In this revised version of the manuscript, the authors describe a novel pathway in melanocytes originating at the Golgi and contributing to melanosome maturation. The authors provide evidence of the role of Rab6 and the adaptor protein ELKS in mediating this pathway. This is a significantly improved version of the original manuscript, in which I previously raised several concerns pertaining limitations of the chosen experimental model, subjective interpretation of some results, and insufficient experimental evidence to support the proposed model. The authors addressed satisfactorily all my concerns by providing additional evidence that validates the model in different primary cell cultures (human and murine) and transgenic mice, by providing quantifications to eliminate subjectivity, and by performing additional experiments to better support the proposed working model. In particular, the inclusion of an ELKS-DCT mutant that does not bind Rab6, provides a link between the proposed pathway and melanosome maturation that was vague before. In general I consider that all my concerns were addressed and I think the current version has been much improved. I do have one minor request to help strengthen the presented data: Please provide the quantification of the results shown in figure 6c with the associated standard error and statistical significance.

Reviewer #2:

Remarks to the Author:

The authors have adequately addressed my comments and I support publication of this paper.

Reviewer #3:

Remarks to the Author:

The authors have satisfactorily addressed my comments and, in my opinion, those of the other reviewers. The authors should be commended for their thorough revision of the manuscript. I would change the title to "The Rab6 secretory pathway contributes to melanosome biogenesis and function".

Reviewer #4:

Remarks to the Author:

I have reviewed the revised version of the paper entitled "Routing of the Rab6 secretory pathway towards the lysosome related organelle of melanocytes" by Patwardhan et al. The authors have performed additional experiments, modified the text and addressed the vast majority of concerns raised in the original version. As a result, the manuscript is significantly improved. I only have relatively minor comments that do not require additional experimentation but are important to accomplish optimal clarity.

While the data shows Rab6/ELKS function in melanosome biogenesis, it does not demonstrate a direct Golgi-melanosome route. Likewise, the data does not show the Rab6 vesicles fuse with the melanosome. Therefore, the abstract should be modified to more accurately reflect the findings and not overstate them. Along the same lines, the model (Figure 6h) should be eliminated because alternative routes/models would also fit the data.

Regarding the following Abstract sentence: "Our data together reveal for the first time that the secretory pathway can be directed towards intracellular organelles to ensure their biogenesis and function" still does not feel accurate. According to the authors' explanation of the idea they are

intending to convey, I suggest the following change: "Our data together reveal for the first time that the secretory pathway can be directed towards intracellular organelles of endosomal origin to ensure their biogenesis and function".

Reviewers' comments:

Reviewer #1 (Remarks to the Author):

In this revised version of the manuscript, the authors describe a novel pathway in melanocytes originating at the Golgi and contributing to melanosome maturation. The authors provide evidence of the role of Rab6 and the adaptor protein ELKS in mediating this pathway. This is a significantly improved version of the original manuscript, in which I previously raised several concerns pertaining limitations of the chosen experimental model, subjective interpretation of some results, and insufficient experimental evidence to support the proposed model. The authors addressed satisfactorily all my concerns by providing additional evidence that validates the model in different primary cell cultures (human and murine) and transgenic mice, by providing quantifications to eliminate subjectivity, and by performing additional experiments to better support the proposed working model. In particular, the inclusion of an ELKS-DCt mutant that does not bind Rab6, provides a link between the proposed pathway and melanosome maturation that was vague before. In general I consider that all my concerns were addressed and I think the current version has been much improved. I do have one minor request to help strengthen the presented data: Please provide the quantification of the results shown in figure 6c with the associated standard error and statistical significance.

We thank the reviewer for having evaluated the revised manuscript. We are glad that the concerns raised by the reviewers were satisfactorily answered. We have now added the associated standard error of the mean and statistical significance (using unpaired t test with equal SD) associated to the quantification of the results shown in Fig6C.

MART-1 and TYRP2 expression associated with melanosomes were decreased about ($55.1 \pm 14\%$ and $45.6 \pm 14.5\%$, associated with *P* values 0.0327 and 0.0197, respectively) in RAB6-depleted melanocytes (see page 11 of the revised manuscript).

Reviewer #2 (Remarks to the Author)

The authors have adequately addressed my comments and I support publication of this paper.

We are glad that the concerns raised by the reviewers were satisfactorily answered.

Reviewer #3 (Remarks to the Author)

The authors have satisfactorily addressed my comments and, in my opinion, those of the other reviewers. The authors should be commended for their thorough revision of the manuscript. I would change the title to "The Rab6 secretory pathway contributes to melanosome biogenesis and function".

We thank the reviewer for having taken the time to revise our manuscript. His/ Her comment is really rewarding.

However we would like to keep the original title in order to support the idea that the RAB6 secretory pathway is not only transported to the plasma membrane, but also to the LRO of melanocytes (aka melanosome). We believe that the fact that the secretory pathway targets directly intracellular organelles is of real importance not only for LRO-producing cells, but more generally for the cell biology and intracellular trafficking fields.

Reviewer #4 (Remarks to the Author)

I have reviewed the revised version of the paper entitled “Routing of the Rab6 secretory pathway towards the lysosome related organelle of melanocytes” by Patwardhan et al. The authors have performed additional experiments, modified the text and addressed the vast majority of concerns raised in the original version. As a result, the manuscript is significantly improved. I only have relatively minor comments that do not require additional experimentation but are important to accomplish optimal clarity.

We thank the reviewer for taking his/her valuable time to examine the revised manuscript. We are glad that the concerns raised by the reviewers were satisfactorily answered.

While the data shows Rab6/ELKS function in melanosome biogenesis, it does not demonstrate a direct Golgi-melanosome route.

We regret but we believe that our data support the role of RAB6/ELKS in a direct Golgi-melanosome route. Especially, our live cell imaging showed that RAB6-derived vesicles emerge from the Golgi area and are directly transported towards melanosomes (Supplementary Movie 3). ELKS functions in the stable association (docking) of RAB6 vesicles with melanosomes (Figure 5), and RAB6 and ELKS associates with melanosomal membranes (Figure 1+4). Several cargos (secretory (NPY) or melanosomal ones (MART-1, TYRP2)) were less associated with melanosomes in absence of RAB6 expression. Hence, we do not believe that our conclusions were overstated

Likewise, the data does not show the Rab6 vesicles fuse with the melanosome. Therefore, the abstract should be modified to more accurately reflect the findings and not overstate them.

We agree with the reviewer that we have provided indirect evidences for the fusion of RAB6-post-Golgi vesicles with melanosomes. Thus we have corrected accordingly the abstract by removing “with which they fuse” in the following sentence “RAB6 and ELKS to directly transport and dock Golgi-derived carriers to melanosomes with which they fuse” (lane 31-33).

Along the same lines, the model (Figure 6h) should be eliminated because alternative routes/models would also fit the data.

Of course, a model is not likely representing the full picture and overall complexity of such pathway. However it reflects the general idea and conclusions driven by the data presented in the manuscript. This model has the advantage to present the current model for the biogenesis of melanosomes to ‘non-pigmentation’ readers and to highlight one of the key point of the paper; the existence of a secretory route that targets intracellular compartment and not only the plasma membrane. Thus if possible, we would like to keep the model because we believe it brings a satisfactory understanding of the main conclusions of the study.

However and in order to highlight that this is a working hypothesis, we have slightly modified the main text (page 16) and associated legend (page 35) by being more cautious with our statements and by mentioning that other alternative routes cannot be excluded.

Regarding the following Abstract sentence: “Our data together reveal for the first time that the secretory pathway can be directed towards intracellular organelles to ensure their biogenesis and function” still does not feel accurate. According to the authors’ explanation of the idea they are intending to convey, I suggest the following change: “Our data together reveal for the first time that the secretory pathway can be directed towards intracellular organelles of endosomal origin to ensure their biogenesis and function”.

We have modified the text accordingly (line 37, page 2).

We would like to thank all the reviewers for their enthusiastic and insightful comments and for the time they have spent to review our manuscript.